

# Mean age from observations in the lowermost stratosphere: an improved method and interhemispheric differences

Thomas Wagenhäuser[1], Markus Jesswein[1], Timo Keber[1], Tanja Schuck[1], Andreas Engel[1]

[1]Institute for Atmospheric and Environmental Sciences, Goethe-University of Frankfurt, Frankfurt, Germany

*Correspondence to*: Thomas Wagenhäuser (wagenhaeuser@iau.uni-frankfurt.de)

**Abstract.** Age of stratospheric air is a concept commonly used to evaluate transport timescales in atmospheric models. The mean age can be derived from observations of a single long-lived trace gas species with a known tropospheric trend. Commonly, deriving mean age is based on the assumption that all air enters the stratosphere through the tropical (TR) tropopause. However, in the lowermost stratosphere (LMS) close to the extra-tropical (exTR) tropopause cross tropopause

transport needs to be taken into account. We introduce the new exTR-TR method, which considers exTR input into the stratosphere in addition to TR input. We apply the exTR-TR method to in situ $SF_6$ measurements from three aircraft campaigns (PGS, WISE and SouthTRAC) and compare results to those from the conventional TR-only method. Using the TR-only method, negative mean age values are derived in the LMS close to the tropopause during the WISE campaign in northern hemispheric (NH) fall 2017. Using the new exTR-TR method instead, the number and extent of negative mean age

values is reduced. With our new exTR-TR method we are thus able to derive more realistic values of typical transport times in the LMS from in situ $SF_6$ measurements. Absolute differences between both methods range from 0.3 to 0.4 years among the three campaigns. Interhemispheric differences in mean age are found when comparing seasonally overlapping campaign phases from the PGS and the SouthTRAC campaigns. On average, within the lowest 65 K potential temperature above the tropopause the NH LMS is 0.5 years ± 0.3 years older around March 2016 than the southern hemispheric (SH) LMS around

September 2019.  The derived differences between results from the exTR-TR method and the TR-only method, as well as interhemispheric differences are higher than the sensitivities of the exTR-TR method to parameter uncertainties, which are estimated to be below 0.22 years for all three campaigns.

## 1 Introduction

The lowermost stratosphere (LMS) is the lowest part of the extra tropical (exTR) stratosphere. Its upper boundary usually is

defined as the 380 K isentrope, which approximates the lower boundary of the stratosphere in the tropics. The chemical composition of the LMS plays an important role in the climate system. Different transport paths and timescales determine the chemical composition of the LMS for a wide range of trace gases. The most prominent transport mechanism in the stratosphere is the Brewer-Dobson circulation (BDC), which transports air from the tropical (TR) tropopause to the exTR and polar stratosphere (Butchart, 2014). The residual circulation part of the BDC is characterized by two branches (Birner





and Bönisch, 2011; Plumb, 2002): One branch extends deep into the middle atmosphere and slowly transports air to high latitudes, where it eventually descends to lower altitudes. The shallow branch in the lower part of the stratosphere transports air poleward below the subtropical transport barrier and is characterized by comparably fast transport time scales. In addition to residual transport, air is transported within the stratosphere by bidirectional mixing. Both residual transport and mixing are induced by wave activity on different scales and are part of the BDC. In addition to the BDC, exTR cross-tropopause

transport strongly affects the chemical composition of the LMS. This exTR transport mechanism is modulated by the subtropical jet (Gettelman et al., 2011).

Age of air is a widely used concept to describe tracer transport in the stratosphere (Waugh and Hall, 2002). In principle, infinitesimal fluid elements enter the stratosphere across a source region. The transit time (or "age") of each individual fluid element is the elapsed time since it last made contact to a source region. A macroscopic air parcel in the stratosphere consists

of an infinite number of such fluid elements, each with its own transit time. The transit time distribution for the air parcel is called the "age spectrum" (Kida, 1983). Past studies were able to obtain information on age spectra from observations of multiple trace gases (Andrews et al., 1999, 2001; Bönisch et al., 2009; Hauck et al., 2020; Ray et al., 2022). The first moment of the age spectrum is the mean age of air. It can be derived from measurements of a single inert trace gas species with a monotonic trend in the troposphere. $CO_2$, $SF_6$ and a variety of "new" age tracers have been used in past studies to

derive the mean age of air from observations (e.g. Engel et al., 2017; Leedham Elvidge et al., 2018).

Age of air from observations provides a stringent test for numerical models. The number of available trace gas observations that are suited to derive mean age is vastly higher than that to derive age spectra. In addition, deriving mean age relies on making less assumptions than does deriving age spectra. This makes mean age a valuable measure to compare models to observations. Still, observational estimates of mean age rely on several simplified assumptions, depending on the trace gas

used, which significantly add to the uncertainty in mean age across large areas of the stratosphere. For example, in order to derive mean age from $SF_6$ measurements, commonly an infinite lifetime is assumed. In contrast, recent studies showed that a mesospheric sink of $SF_6$ leads to a significant bias towards higher ages especially on old mean age values derived from $SF_6$ observations (Leedham Elvidge et al., 2018; Loeffel et al., 2022). Another common assumption is that all air enters the stratosphere through the TR tropopause. However, Hauck et al. (2019, 2020) showed that in the vicinity of the tropopause,

transport across the exTR tropopause is also important to adequately describe age spectra and mean age in the LMS. While the assumption of a single entry point is a good approximation for the stratosphere above about 380 K potential temperature θ, this is thus not the case for the LMS. Together with the interhemispheric gradient in tropospheric trace gas mixing ratios, this limits the ability to derive mean age of air in the LMS. Further improvements of the methods to derive the mean age of air from observations are thus desirable in order to provide robust real world estimates of transport time scales in sensitive

regions of the atmosphere and be able to compare to model results.

With this work we focus on mean age of air in the LMS, where the old bias of $SF_6$ mean age is presumably low. We introduce an extended method that considers exTR input into the stratosphere in addition to TR input (hereafter exTR-TR method). In Sect. 2 we describe the concept and implementation of our new exTR-TR method. In Sect. 3 firstly we compare



results from the exTR-TR method to the conventional method, which only considers TR input (hereafter referred to as
TR-only method). These results are based on in situ measurements taken during three aircraft campaigns. Secondly, we
compare northern hemispheric (NH) and southern hemispheric (SH) mean age in the LMS based on these results. Thirdly,
we present a sensitivity study on the exTR-TR method. We summarize our findings in Sect. 4.

## 2 Calculating Mean Age in the LMS considering multiple entry regions

### 2.1 General concept

A common approach to describe the mixing ratio $\chi(\boldsymbol{x})$ of a suitable age tracer at an arbitrary location $\boldsymbol{x}$ in the stratosphere is

$$\chi(\boldsymbol{x}) = \int_0^\infty \chi(\boldsymbol{x_0}, t') * G(\boldsymbol{x}, t') dt', \tag{1}$$

with $\chi(\boldsymbol{x_0}, t')$ being the tracer mixing ratio time series in the source region $\boldsymbol{x_0}$ as a function of transit time $t'$ and the age
spectrum $G(\boldsymbol{x}, t')$. The approach expressed in Eq. (1) is based on the assumption, that all fluid elements that enter the
stratosphere at the same time have the same tracer mixing ratio. Albeit, in the real world there is no suitable age tracer with
the same mixing ratio time series throughout the troposphere. Hence, the mixing ratio time series is likely to be different in
different entry regions. By using Eq. (1), so far studies that derived the mean age of stratospheric air from measurements of
one inert trace gas commonly relied on the assumption, that all air enters the stratosphere through the tropical (TR)
tropopause (TR-only method), which appears valid for large parts of the stratosphere. In the LMS however, exTR input
needs to be considered (Hauck et al., 2019, 2020).

We introduce the new exTR-TR method, which builds on an extended approach to derive mean age in the LMS from an inert
monotonic tracer that considers exTR input into the stratosphere in addition to TR input. In a generalized way, our extended
approach accounts for input into the stratosphere from $N$ individual source regions $\boldsymbol{x_i}$ with individual mixing ratio time series
$\chi(\boldsymbol{x_i}, t')$ by calculating a weighted mixing ratio time series. We use the origin fractions $f_i(\boldsymbol{x})$ as introduced by Hauck et al.
(2020) as weights for each $\chi(\boldsymbol{x_i})$. $f_i(\boldsymbol{x})$ is the fraction of air at $\boldsymbol{x}$, that entered the stratosphere through $\boldsymbol{x_i}$. By applying this
assumption, Eq. (1) translates into Eq. (2):

$$\chi(\boldsymbol{x}) = \int_0^\infty \sum_{i=0}^{N-1} \big(f_i(\boldsymbol{x}) * \chi(\boldsymbol{x_i}, t')\big) * G(\boldsymbol{x}, t') dt'$$

$$= \sum_{i=0}^{N-1} \big(f_i(\boldsymbol{x}) * \int_0^\infty \chi(\boldsymbol{x_i}, t') * G(\boldsymbol{x}, t') dt'\big) \tag{2}$$

Note that Eq. (2) is only valid if the sum of all origin fractions equals 1:

$$\sum_{i=0}^{N-1} f_i(\boldsymbol{x}) := 1 \tag{3}$$

There are currently no long term time series from measurements at the tropopause that are suited for mean age calculations.
For this reason, we assume that each long term time series at each entry region i can be described by the tropical ground time
series shifted by an individual constant $t_{xi}$:

$$\chi(\boldsymbol{x_i}) = \chi(\boldsymbol{x_{TR\ ground}}, t' - t_{xi}) \tag{4}$$





The negative sign points out, that looking at increasing transit times means looking backwards in time. In case of an ideal

inert linear tracer with the y-intercept $a$ and slope $b$ and by applying Eq. (4), Eq. (2) can be transferred to Eq. (5) in order to

calculate the mean age $\Gamma(\boldsymbol{x})$:

$$\Gamma(\boldsymbol{x}) = \frac{a - \chi(x)}{b} + t_m(\boldsymbol{x}), \tag{5}$$

with the weighted mean time shift $t_m(\boldsymbol{x}) = \sum_{i=0}^{N-1}(f_i(\boldsymbol{x}) * t_{xi})$.

In case of an ideal inert quadratic tracer with curvature $c$ and a known ratio of moments $\lambda = \frac{\Delta^2}{\Gamma}$ with the width of the age

spectrum $\Delta$ and again by applying Eq. (4), Eq. (2) can be transferred to Eq. (6) in order to calculate mean age:

$$\Gamma(\boldsymbol{x})_{1,2} = -\lambda + t_m(\boldsymbol{x}) + \frac{b}{2c} \pm \sqrt{\left(-\lambda + t_m(\boldsymbol{x}) + \frac{b}{2c}\right)^2 - \frac{a + bt_m(\boldsymbol{x}) - \chi(x)}{c} - \sum_{i=0}^{N-1}[f_i(\boldsymbol{x}) * t_{xi}^2]}. \tag{6}$$

Details on deriving Eq. (5) and Eq. (6) are given in the Appendix A. Obviously, Eq. (5) and Eq. (6) can also be applied to the

single entry region case, i.e. in context of the conventional TR-only method. This is equivalent to deriving mean age from an

ideal inert linear tracer following Hall and Plumb (1994), respectively in the quadratic case following Volk et al. (1997).

Alternatively, instead of assuming ideal linearly or ideal quadraticly evolving tracer mixing ratios, $G$ can be approximated by

a mathematical function, e.g. an inverse Gaussian following (Hall and Plumb, 1994). Information on the width of the age

spectrum needs to be included (like in the ideal quadratic tracer case). This way, Eq. (1) (TR-only) or Eq. (2) (exTR-TR) can

be directly used to create a lookup table for $\Gamma$ from a range of age spectra $G$ as described in several studies (e.g. Fritsch et al.,

2020; Leedham Elvidge et al., 2018; Ray et al., 2017). Mean age then is inferred from the best match between measured

$\chi(\boldsymbol{x})$ and mixing ratios given in the lookup table. We refer to this approach as G-match approach in the following.

Our exTR-TR method will only work for inert monotonic tracers, e.g. $SF_6$-like tracers. Tracers that are characterized by

seasonally varying trends in their mixing ratios, which propagate into the stratosphere, e.g. $CO_2$-like tracers, will lead to

ambiguous mean age results in the LMS using the exTR-TR method. We tested calculating mean age from $SF_6$

measurements using Eq. (6) versus following the G-match approach and found only negligible differences for mean ages

greater than one year. For lower mean ages the G-match approach leads to numerical issues that cause larger deviations.

Therefore, we decided to use Eq. (6) for all mean age calculations in context of this study.

### 2.2 Implementation

The new exTR-TR method requires additional information compared to the conventional TR-only method. In order to

account for input from different entry regions, firstly information on the fraction of air that originated from each entry region

is essential. Secondly, the age tracer's mixing ratio time series at each entry region needs to be known. In the following we

introduce a parameterization of the origin fractions published in Hauck et al. (2020). Further, we derive entry mixing ratio

time series by shifting the tropical ground mixing ratio time series by a constant amount of time. The software

implementation of the exTR-TR method is described in the supplementary information.



### 2.2.1 Parameterizations of origin fractions from CLaMS

Information on the fraction of air originating from different source regions is an essential input to our new exTR-TR method. We use the seasonally averaged origin fractions from the Chemical Lagrangian Model of the Stratosphere (CLaMS, e.g. Pommrich et al. (2014) published in Hauck et al. (2020). Hauck et al. (2020) derived such fractions based on origin tracers initiated at three tropopause sections in the model for extra tropical input from the Southern Hemisphere (30 to 90 °S, hereafter referred to as SH input), tropical input (30 °S to 30 °N, hereafter referred to as TR input) and extra tropical input

from the Northern Hemisphere (30 to 90 °N, hereafter referred to as NH input). In total, there are 15 seasonal distributions of origin fractions $f_{i,seas}(x)$ published in Hauck et al. (2020) (see also their Fig. 2): Five seasonal sets (annual mean (ANN), December/January/February (DJF), March/April/May (MAM), June/July/August (JJA) and September/October/November (SON)) for each entry region (SH, TR, NH). (Hauck et al., 2020) found that cross-hemispheric transport is negligible, with origin fractions below 10 % from the extra tropics of the respective other hemisphere. Hence, in order to calculate the mean

age at a given location in the stratosphere, we only consider the exTR origin fraction of the respective hemisphere and assume that the rest originates from the TR tropopause (i.e. $f_{TR} = 1 - f_{exTR}$). By doing so, the number of seasonal distributions of origin fractions reduces from 15 to 10.

In order to facilitate accessing the origin fractions from Hauck et al. (2020) and to reduce computational effort we designed a general mathematical parameterization function $\varphi_{i,seas}$ with 12 parameters to derive 2-D parameterizations for exTR origin

fractions. The process of designing $\varphi_{i,seas}$ was guided by a non-physical but entirely geometrical approach. We chose the potential temperature difference to the local 2 PVU tropopause (ΔΘ) as the vertical coordinate, equivalent latitude (eq. lat., i.e. latitudes sorted by potential vorticity) as the horizontal coordinate. Details on the parameterizations and on how we derived them are given in the Appendix B. Figure 1 shows $\varphi_{i,seas}$ (top row), $f_{i,seas}(x)$ (middle row) and the absolute difference between $f_{i,seas}(x)$ and $\varphi_{i,seas}$ (bottom row) exemplarily for NH spring (March, April, May: MAM; left column),

NH fall (September, October, November: SON; middle column) and SH spring (SON; right column). The remaining seven distributions are presented in the same way as Fig. 1 in the Supplementary Information (Fig. S1). The absolute differences between $\varphi_{i,seas}$ and $f_{i,seas}(x)$ shown in Fig. 1 are less than 10 % for NH MAM (panel (g)) and SH SON (panel (i)) and only exceed 10 % in a small region at the equator around 25 K above the tropopause for NH SON (panel (h)). The root mean squared difference (RMSD) is less than 3 % for all distributions shown in Fig. 1 and less than 4 % for all 10 distributions

(including the seven distributions shown in Fig. S1).



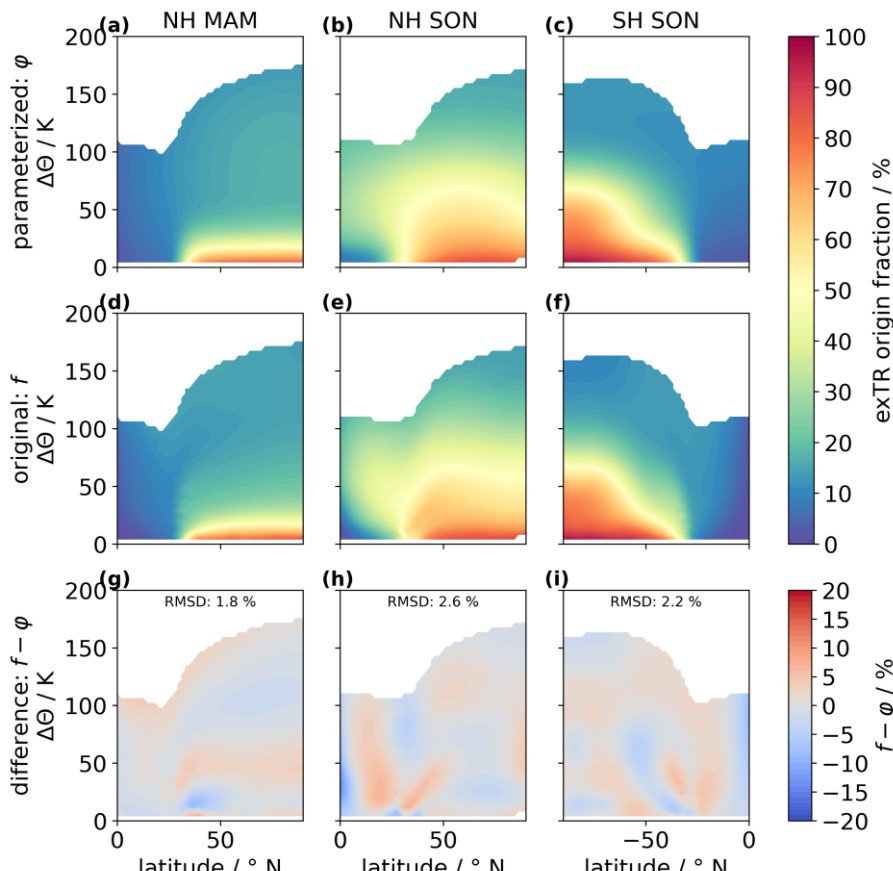

**Figure 1.** Comparing seasonal hemispheric extra tropical origin fractions parameterization ($\varphi$, top row) and original CLaMS fractions from simulations from Hauck et al. (2020) ($f$, middle row). The difference is shown in the bottom row. Only three selected season-hemisphere combinations are shown here (NH MAM, left column; NH SON, middle column; SH SON, right column). The remaining distributions are shown in Fig. S1. Vertical coordinates are given in potential temperature above the 2 PVU tropopause ($\Delta\theta$).

### 2.2.2 Entry region mixing ratio time series

Our new exTR-TR method uses a TR ground reference time series $\chi(x_{TR\ ground}, t')$ together with constant time shift values $t_{xi}$ in order to simulate reference time series in the three entry regions as defined by the origin fractions from Hauck et al. (2020) (see Eq. 4). This will work for inert monotonic tracers like $SF_6$. In contrast, the entry region mixing ratio time series of tracers like $CO_2$, which are characterized by a pronounced seasonality in the troposphere, most likely cannot be approximated satisfactorily with this approach. These tracers are thus not suited for deriving mean age with the exTR-TR method in the LMS. Here, at first we describe, which TR ground reference time series we use and secondly how we derived constant time shift values $t_{xi}$.

In this study, we use $SF_6$ as an age tracer. Simmonds et al. (2020) used ground measurements from the AGAGE (Advanced Global Atmospheric Gases Experiment) Network (Prinn et al., 2018) together with measurements of archived air samples and the two-dimensional AGAGE 12-box model (Cunnold et al., 1978, 1983; Rigby et al., 2013) to derive a monthly



resolved time series of $SF_6$ mixing ratios from the 1970s to 2018. We use the TR ground $SF_6$ mixing ratios of an updated version of this dataset, which has been extended until the end of 2019, as a reference time series $\chi(x_{TR\ ground}, t')$ for calculating the mean age of air.

In order to derive $t_{xi}$ for each of the three entry regions we use the annual mean optimized three-dimensional $SF_6$ mixing ratios output from the Model for Ozone and Related Tracers (MOZART v4.5) for 1970 to 2008 published by Rigby et al. (2010, Supplement). Rigby et al. (2010) derived a new estimate of $SF_6$ emissions using the Emissions Database for Global Atmospheric Research (EDGAR v4) as a prior and optimizing the emissions using $SF_6$ ground measurements from the AGAGE Network including monitoring site data and archived sample measurements together with MOZART and

meteorological data from the National Centers for Environmental Prediction/National Center for Atmospheric Reseach (NCEP/NCAR) reanalysis project. The annually-averaged, three-dimensional optimized $SF_6$ mixing ratio fields that we use to derive $t_{xi}$ are part of their result. In our approach, we only considered the data from 1973 to 2008, since the data from 1970 to 1972 may be influenced by the start conditions of the model (Rigby et al., 2010). We calculated $t_{xi}$ for each of the three entry regions following three steps:

(i) Calculate a mean TR ground $SF_6$ time series by using MOZART data between -30° to 30° N weighted by latitude.

(ii) For each grid cell and each year of the three-dimensional $SF_6$ field time series interpolate $SF_6$ mixing ratios to TR ground time using (i) and calculate time shift to TR ground.

(iii) For each entry region, calculate mean and standard deviations weighted by latitude and pressure for time shifts from (ii) for 1973 to 2008 altogether to eventually obtain $t_{xi}$ and information on associated uncertainty.

The latitudinal extents of the entry regions that we calculated $t_{xi}$ for are the same as for the origin fractions by Hauck et al. (2020). For the exTR entry regions, we included data between 500 hPa and 200 hPa. For the TR entry region, we included data between 300 hPa and 100 hPa. Table 1 lists $t_{xi}$ and standard deviations for the three entry regions SH exTR, TR and NH exTR. Positive values of $t_{xi}$ indicate, that the corresponding region lags behind TR ground $SF_6$ mixing ratios. Note that for NH exTR $t_{xi}$ is negative. This means that this region precedes TR ground $SF_6$ mixing ratios. This finding is consistent

with $SF_6$ source regions being located primarily in the northern hemisphere (Rigby et al., 2010).

We performed a Monte-Carlo simulation in order to test if $t_{xi}$ can be considered to be constant over time for each entry region. Firstly, for each entry region we calculated weighted means and standard deviations for each year (instead for the whole time period like in (iii)). Figure 2 shows the resulting time shift time series from 1973 to 2008. Secondly, for each year, we took 10000 samples from a Gaussian distribution using those weighted means and standard deviations in order to

create 10000 time series for each entry region. Thirdly, we applied a linear fit to each of the 10000 time series and calculated the mean and the standard deviation of the slope for each entry region. The resulting mean slopes, standard deviations and the ratio of mean slope and standard deviation are listed in Table 2. For NH exTR and TR the mean slopes deviate less than one standard deviation from zero. For SH exTR the mean slope deviates less than 1.2 standard deviations from zero. Hence, we do not detect a significant trend. These findings strengthen our confidence into our assumption that we can use the



constant time shifts $t_{xi}$ listed in Table 1 together with $\chi(x_{TR\ ground}, t')$ to describe the SF$_6$ entry mixing ratio time series

reasonably well. In context of our exTR-TR method we assume that this also holds true for the subsequent decade from 2008

on. This decade is not covered by the model from Rigby et al. (2010) that we used to derive $t_{xi}$, however it is covered by

$\chi(x_{TR\ ground}, t')$ (updated from Simmonds et al., 2020). We emphasize that each $t_{xi}$ as defined here is an integrated

empirical measure. $t_{xi}$ does neither contain useful information on transport paths nor on transit times from the TR ground to

the entry regions. We only use $t_{xi}$ to derive entry mixing ratio time series at locations, where suitable long term time series

are not available from measurements.

**Table 1:** $t_{xi}$ and standard deviations for SF$_6$ mixing ratio time series at three entry regions (NH 30°N-90°N, 500-200 hPa; TR 30°S-30°N, 300-100 hPa; SH 30°S-90°S, 500-200 hPa), weighted by latitude and pressure for the time period 1973-2008. These time shifts have been calculated relative to TR ground.

|  | $t_{xi}$ / years | weighted standard deviation / years |
|---|---|---|
| NH exTR entry region | -0.4 | 0.156 |
| TR entry region | 0.12 | 0.15 |
| SH exTR entry region | 0.53 | 0.074 |


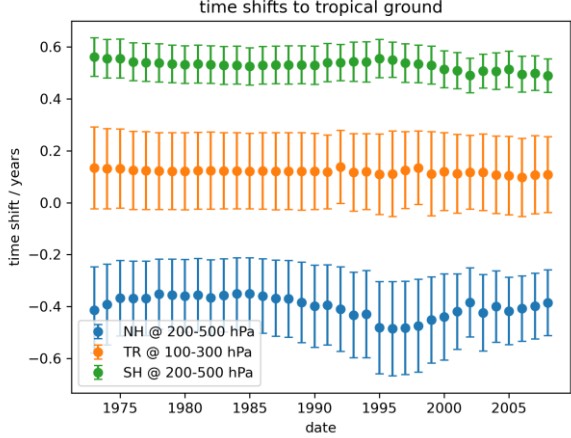

**Figure 2.** Annual mean time shifts and standard deviations for SF$_6$ mixing ratio time series at three entry regions, weighted by latitude and pressure. Based on three dimensional annual SF$_6$ model data output from Rigby et al. (2010).



**Table 2:** Mean slopes and slope standard deviation from Monte-Carlo simulation using the data shown in Fig. 2. Ratios of mean slopes
and standard deviations have been calculated prior to rounding.

| | mean time shift slope / years years-1 | time shift slope standard deviation / years years-1 | ratio of mean slope and standard deviation |
|---|---|---|---|
| NH exTR entry region | $-2*10^{-3}$ | $3*10^{-3}$ | $-0.9$ |
| TR entry region | $-5*10^{-4}$ | $3*10^{-3}$ | $-0.2$ |
| SH exTR entry region | $-1*10^{-3}$ | $2*10^{-3}$ | $-1.1$ |

## 2.3 Stratospheric observations of age tracer SF$_6$

We apply our new exTR-TR method to in situ measurements of SF$_6$ that were obtained during three HALO research
campaigns. The first campaign, PGS (Oelhaf et al., 2019), is a combination of three missions: POLSTRACC (Polar
Stratosphere in a Changing Climate), GW-LCYCLE (Investigation of the Life cycle of gravity waves) and SALSA
(Seasonality of Air mass transport and origin in the Lowermost Stratosphere). PGS was split into two campaign phases, that
were conducted in NH winter 2015/16 between December 13 and February 2, respectively NH early spring 2016 between
February 26 and March 18. The second campaign, WISE (Wave-driven Isentropic Exchange, https://www.wise2017.de, last
access: 05 May 2022), took place mainly in NH fall 2017 between September and October. The flight tracks of HALO
during the PGS and during the WISE campaign are shown in Keber et al. (2020) Fig. 2 and Fig. 1b. They cover large parts of
mid- and high latitudes in the NH. Thirdly, we consider data from the SouthTRAC (Southern Hemisphere Transport,
Dynamics and Chemistry) campaign, which took place in SH spring 2019. The SouthTRAC campaign also was split into two
campaign phases, conducted between September 6 and October 9 (Rapp et al., 2021), respectively between November 2 and
November 15. The flight tracks for SouthTRAC, which covered a wide geographical area of the SH, are shown in Jesswein
et al. (2021).

SF$_6$ and CFC-12 measurements were obtained in-flight in context of all three campaigns with a time resolution of one minute
using the ECD channel of the two-channel Gas chromatograph for Observational Studies using Tracers (GhOST) instrument
in a similar set-up as used in the SPURT campaign (Bönisch et al., 2009; Engel et al., 2006). SF$_6$ has been measured with a
precision of 0.6 % (0.56 %) during SouthTRAC (PGS, WISE). CFC-12 has been measured with a precision of 0.23 %
(0.2 %) during SouthTRAC (PGS, WISE). All measurements are reported relative to the AGAGE SIO-05 scale (Miller et al.,
2008; Prinn et al., 2018; Rigby et al., 2010; Simmonds et al., 2020). Due to the better precision of CFC-12 measurements,
the original SF$_6$ data was smoothed using a local SF$_6$ – CFC-12 correlation 10 minutes before and after each measurement
following Krause et al. (2018), prior to calculating mean age values. The height of the dynamical 2 PVU tropopause (e.g.
Gettelman et al., 2011) as well as eq. lat. coordinates were obtained via CLaMS driven by ERA-5 reanalysis along the flight
tracks. With this study we exclusively focus on the LMS. Therefore, only tracer measurements at or above the dynamical
tropopause were considered (i.e. with $\Delta\theta \geq 0\,K$). Tracer measurements and meteorological data are accessible via the
HALO database (HALO consortium, 2021).



## 3 Results and discussion

We derive mean age in the LMS using in situ $SF_6$ measurements from three aircraft campaigns (see Sect. 2.3). Results are presented in a two dimensional tropopause-relative coordinate system. The potential temperature relative to the local

dynamical tropopause (defined by the value of 2 PVU) $\Delta\theta$ is used as vertical coordinate. Horizontally, data are sorted by eq. lat. In order to visualize and compare our results, datasets were processed in a three-step process:

1. Mean age was calculated for each data point that was measured above the local 2 PVU tropopause.
2. For each campaign dataset, mean ages were averaged in $\Delta\theta$ – eq. lat. bins (5 K – 5 °). Only bins that contained at least five data points were considered.

3. The averaged mean ages have been corrected for mesospheric loss using a linear correction function by (Leedham Elvidge et al., 2018):

$$\Gamma_{corr} = 0.85 * \Gamma - 0.02 \; years \tag{7}$$

### 3.1 Method comparison using campaign averaged results

We applied our new exTR-TR method for deriving mean age in the LMS considering exTR and TR input into the

stratosphere to all three campaign datasets. Further, we applied the conventional TR-only method, which considers only TR input into the stratosphere, in order to compare the results from both methods. The results were averaged and corrected for mesospheric loss.

Figure 3 shows the resulting $\Delta\theta$ – eq. lat. distributions of averaged mean age mA for PGS (left column), WISE (middle column) and SouthTRAC (right column), derived using the conventional TR-only method $mA_{TR-only}$ (top row), using our new

exTR-TR method $mA_{exTR-TR}$ (middle row), and the difference between the two methods $\Delta mA_{methods}$ (bottom row). There are negative values down to -0.54 years close to the tropopause below $\Delta\theta=10$ K in the WISE dataset using the TR-only method (panel (b)). In the same region mean ages between -0.23 and 0.35 years are found using the new exTR-TR method (panel (e)). Mean ages below 0 as derived from the TR-only method do not allow for a reasonable interpretation regarding transport time scales in the LMS. In contrast, mean ages derived using our new exTR-TR method appear physically reasonable even

close to the tropopause. During the WISE campaign low gradients in $mA_{exTR-TR}$ values reveal a well-mixed LMS (panel (e)), while during PGS and ST stronger gradients in $mA_{exTR-TR}$ are found (panels (d) and (f)).

The maximum absolute difference between average exTR-TR and TR-only method derived mean ages $|\Delta mA_{methods}|$ is with 0.31 years (WISE and PGS) and 0.42 years (SouthTRAC) on the same order of magnitude for all three campaigns, but in different direction for the SH (see also Fig. 3, bottom row). For all three campaigns $|\Delta mA_{methods}|$ is largest close to the

tropopause at mid- and high latitudes and approaches zero years further up and closer to the equator. This distribution is similar to the distribution of exTR origin fractions from CLaMS. In fact, $|\Delta mA_{methods}|$ and the exTR origin fractions are highly correlated (r > 0.99 for all three campaigns). This results from the design of the exTR-TR method, which explicitly considers exTR input into the stratosphere.




Note that in the SH data from the SouthTRAC campaign mean ages are generally lower when derived using the exTR-TR

method than when using the TR-only method. For NH data (WISE, PGS) the opposite is the case. This is a direct

consequence from the TR ground mixing ratio time series being lagged by a positive (in the SH), respectively a negative (in

the NH) empirical time shift to obtain the respective entry mixing ratio time series (see Table 1).

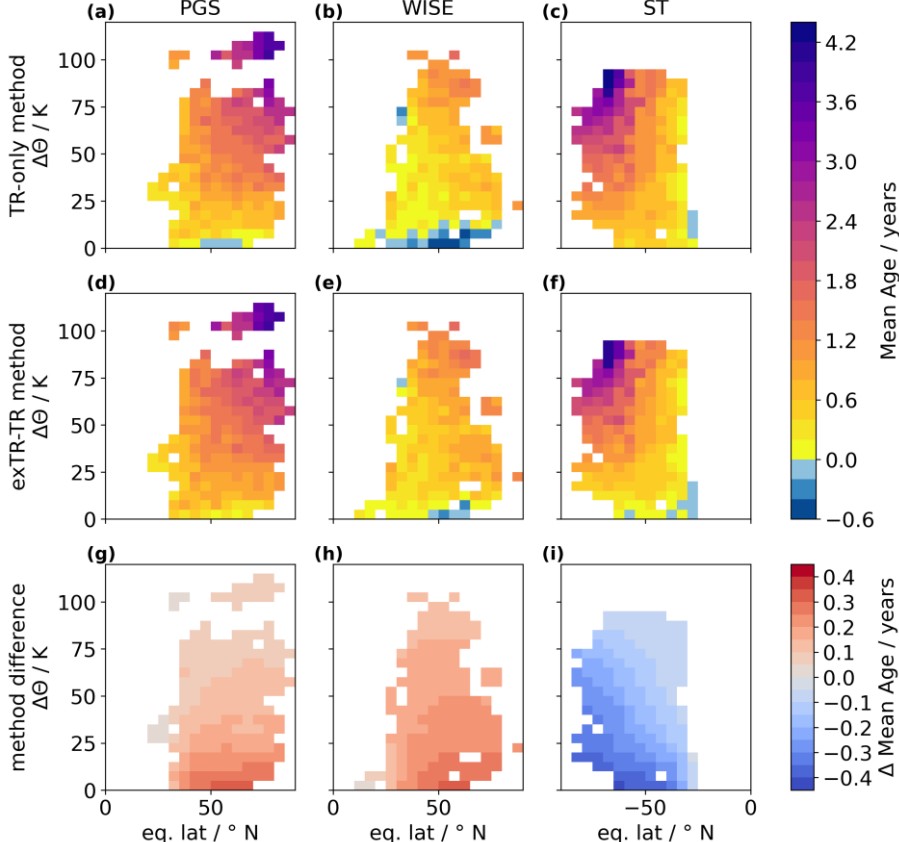

**Figure 3.** Comparison of methods to derive mean age in the LMS during PGS (left column), WISE (middle column) and SouthTRAC
(right column). Results are shown for the TR-only method (top row) and the new exTR-TR method (middle row). The absolute differences
between both methods are shown in the bottom row. Altitude is given in Δθ. Horizontally, data are sorted by eq. lat.

### 3.2 SouthTRAC and PGS campaign: SH vs NH late winter/early spring

Both the SouthTRAC and the PGS campaign involved flights during the respective hemisphere's late winter /early spring.

We compare results from the SouthTRAC campaign phase 1 dataset (ST1) to results from the PGS campaign phase 2 dataset

(PGS2) derived with the exTR-TR method. This selection is a compromise between including a high number of trace gas

measurements and having a large seasonal overlap between both datasets. Again, the results were averaged and corrected for

mesospheric loss.





Figure 4 shows $mA_{exTR-TR}$ for ST1 (panel (a)), for PGS2 (panel (b)) and the difference between the two $mA_{PGS2-ST1}$ (panel (c)). In order to calculate interhemispheric differences, the $mA_{exTR-TR}$ distribution from ST1 was converted from eq.

lat. degrees North to eq. lat. degrees South by flipping it horizontally. Hence, the respective pole corresponds to 90 ° eq. lat. for both datasets in panel (c). Mean ages $mA_{exTR-TR}$ between -0.2 years and 0.1 years are found within the lowest 10 K above the tropopause during ST1 (panel (a)). During PGS2 $mA_{exTR-TR}$ between 0.2 years and 1.1 years are found in the equivalent region in the NH (panel (b)). The differences in mean age $mA_{PGS2-ST1}$ (panel (c)) reveal higher mean ages during PGS2 than during ST1 from the tropopause up to 65 K above the tropopause throughout mid- and high latitudes, with a few exceptions.

On average, below $\Delta\theta$=65 K the LMS is 0.5 years ± 0.3 years older during PGS2 than during ST1. Above $\Delta\theta$=65 K a more complex picture is observed: At $\Delta\theta$-levels between 65 K and 85 K at latitudes between 40° and 55° the LMS is even older during PGS2 than during ST1 with $mA_{PGS2-ST1}$=0.7 years ± 0.4 years on average. In contrast, the opposite is the case at the same $\Delta\theta$-levels but at poleward latitudes higher than 55°: Mean ages during ST1 are older than during PGS2, with $mA_{PGS2-ST1}$ reaching values down to -2.1 years. A less clear picture emerges when comparing mean ages derived with the

TR-only method (see Appendix C: Fig. C1). This could be explained by the fact, that the TR-only method disregards the interhemispheric gradient in $SF_6$ mixing ratios. In the LMS, the resulting mean age values thus are low biased in the NH, while they are old biased in the SH using the TR-only method. These biases happen to obscure interhemispheric differences in mean age in the LMS which have been detected using the new exTR-TR method on the same dataset.

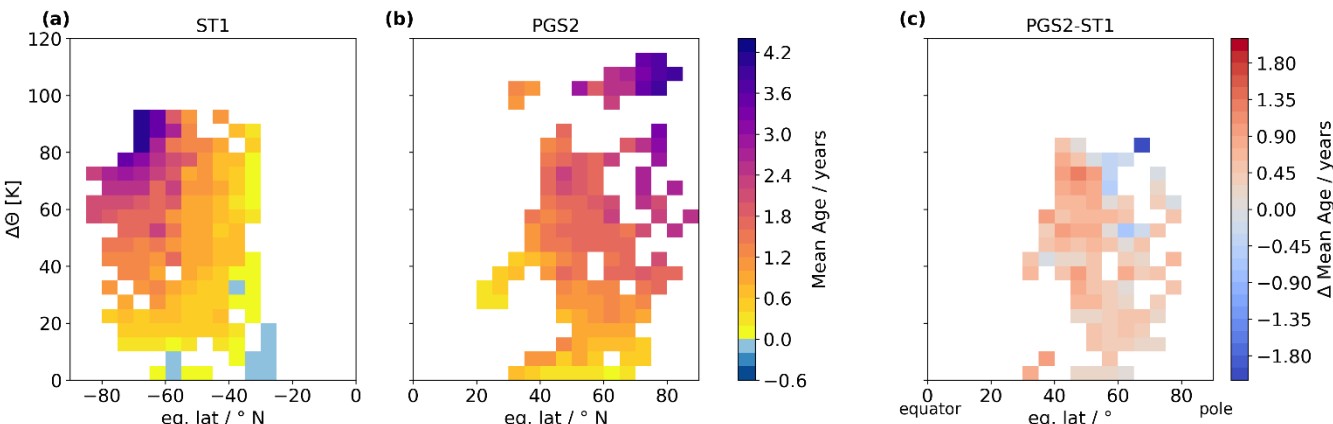

**Figure 4.** Comparison of mean age latitude-altitude distributions during SouthTRAC phase 1 (ST1) (a), PGS phase 2 (PGS2) (b). The hemispheric difference based on these two campaign phases is shown in panel (c). Altitude is given in $\Delta\theta$. Horizontally, data are sorted by eq. lat.

Our findings indicate, that on the one hand, during ST1 old air from higher altitudes descends in a confined way at high latitudes. There is a sharp vortex edge with a strong gradient in the SH. On the other hand, during PGS2 descending old air is

mixed vertically and horizontally with young air in the LMS. The vortex edge is less sharp than during ST1, resulting in younger air at high latitudes and altitudes and older air outside the PGS2 vortex region compared to ST1.

These results cover only isolated time periods of less than two months for each campaign. In addition, as discussed by Jesswein et al. (2021) the extend of the respective polar vortices and therefore also the location of the respective vortex edge





are likely to be different for both hemispheres. Hence, different vortex characteristics contribute to the differences observed

in Fig. 4. Nevertheless, our findings are in agreement with multiannual simulation results from Konopka et al. (2015), who found a pronounced minimum in wave forcing driving the shallow branch of the BDC in the midlatitudes of the lower stratosphere in the SH between June and October, opposed to a maximum in boreal spring in the NH.

### 3.3 Sensitivity study

### 3.3.1 Procedure

Our new exTR-TR method requires input of several parameters, which all have individual uncertainties. In the following, the sensitivity of the exTR-TR method to these uncertainties is investigated in context of three aircraft campaigns.

We identified seven uncertain parameters:

(i) extra tropical origin fraction; $f_{exTR,seas}(\boldsymbol{x})$

(ii) time shift to extra tropical entry region; $t_{exTR}$

(iii) time shift to tropical entry region; $t_{TR}$

(iv) measurement precision of age tracer mixing ratios; $\chi(\boldsymbol{x})$

(v) ratio of moments; $\lambda$

(vi) chemical depletion of age tracer $SF_6$

(vii) reference time series calibration scale uncertainty

Parameters (i) to (vi) may vary for each individual observational sample, making them eligible for a sensitivity analysis. In contrast, the uncertainty of the calibration scale of the reference time series (vii) affects all derived absolute mean ages not in an individual but in a consistent way. Therefore, we excluded it from the sensitivity analysis, albeit knowing that it contributes to the overall uncertainty in deriving mean age from tracers. Furthermore, we excluded the chemical depletion of $SF_6$ (vi) from the sensitivity analysis, since it is not yet well understood and subject of current comprehensive research (e.g.

Loeffel et al., 2022). Leedham Elvidge et al. (2018) showed that younger mean age values derived from $SF_6$ measurements, which point to shorter transport paths, are less effected by the mesospheric sink than are older mean age values. Adequately addressing uncertainties in mean age due to the chemical depletion of $SF_6$ is beyond the scope of this paper, which focuses primarily on young air in the LMS.

Parameters (i) to (v) are suited for a sensitivity analysis within the scope of this study using a Monte-Carlo simulation. Since

typical mean ages and origin fractions vary across different locations and seasons in the LMS, the sensitivity of the exTR-TR method is investigated and results are shown in the same two dimensional tropopause-relative coordinate system that is used for the results shown in Sects. 3.1 and 3.2: $\Delta\theta$ is used as vertical coordinate, while horizontally data are sorted by eq. lat. We conduct the sensitivity analysis by applying the following procedure to each of the three aircraft campaign datasets individually.



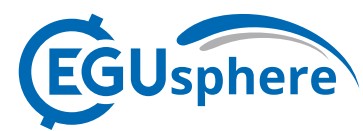

In order to obtain a reduced set of representative data and therefor reduce computational effort of the subsequent steps, SF$_6$ mixing ratios and dates of observation are averaged into 5 K $\Delta\theta$ and 5° eq. lat. bins. For each bin the following three steps are applied:

Step 1: For each uncertain parameter (i)-(v), a random number is drawn based on the parameter's best estimate and its uncertainty for that specific bin (see below for details). This is done 1000 times to create 1000 sets of parameters.

Step 2: These 1000 sets of parameters are used to calculate 1000 mean age values.

Step 3: The standard deviation of those 1000 mean age values is calculated to obtain an overall sensitivity value for this bin.

This way we derive the overall sensitivity. Further, we investigate the relative importance of the uncertain parameters (i)-(v). For this purpose, additional sensitivity calculations are done where only one uncertain parameter is varied while leaving the others at their best estimate.

**3.3.2 Parameter uncertainties**

Here we describe how uncertainties associated with parameters (i)-(v) are implemented in step 1 of the sensitivity analysis.

(i): The exTR origin fraction $f_{exTR}(x)$ varies spatially and over time. For each of the three aircraft campaigns the spatial distribution of uncertainties in $f_{exTR}(x)$ is derived from the parameterized origin fraction $\varphi_{i,seas}(x)$ (see Sect. 2.2.1) individually. Therefore, for each bin the mean absolute half difference (MAHD$_{seas}$) between $\varphi_{i,seas}(x)$ and $\varphi_{i,next\ seas}(x)$,

respectively $\varphi_{i,previous\ seas}(x)$ is calculated. In addition, the root mean squared difference (RMSD$_{space}$) between each bin and its eight surrounding bins is calculated. Both measures, MAHD$_{seas}$ and RMSD$_{space}$ are combined in the root sum squared to finally derive the spatial distribution of uncertainties in the exTR origin fraction for each campaign. Random values are drawn from a Gaussian distribution using this root sum squared.

(ii), (iii): Obtaining $t_{exTR}$ for the NH and for the SH tropopause and $t_{TR}$ for the tropical tropopause regarding SF$_6$ from

annually averaged three dimensional model output is described in Sect. 2.2.2. We use the weighted mean values and standard deviations given in Table 1 as input for a Gaussian distribution, from which random values are drawn.

(iv): The measurement precision of age tracer mixing ratios $\chi(x)$ is given campaign-wise in Sect. 2.3. Since we use smoothed SF$_6$ mixing ratios by considering local CFC-12 – SF$_6$ correlations for mean age calculations, here we apply the better measurement precision for CFC-12 mixing ratios to draw samples from a Gaussian distribution.

(v): Regarding the ratio of moments $\lambda$, random values are drawn from a triangular distribution with a minimum of $\lambda = 0.7$ years, a centre of $\lambda = 1.2$ years and a maximum of $\lambda = 2$ years.

**3.3.3 exTR-TR method sensitivities during PGS, WISE and SouthTRAC**

The sensitivities of the exTR-TR method to uncertainties in input parameters have been calculated following the procedure outlined above. The resulting distributions of sensitivity values are shown in Fig. 5. The most sensitive regions are found

between 20-40 ° poleward of the equator below $\Delta\theta$=20 K during all three aircraft campaigns with maximum values of



0.22 years (PGS, panel (a)), 0.19 years (WISE, panel (b)) and 0.16 years (SouthTRAC, panel (c)). Above Δθ=20 K, the sensitivity values are distributed evenly (standard deviation <0.02 years), with average values of 0.15 years (PGS) and 0.14 years (WISE and SouthTRAC). These sensitivities are lower than the differences between mean ages derived using the exTR-TR method and the TR-only method, which are found to be larger than 0.3 years close to the tropopause.

The contribution of the individual parameters (i)-(v) is shown in Fig. 6. Each row depicts isolated sensitivities to uncertainties in a single parameter with all other parameters being held at their best estimate. This allows us to test the relative importance of the individual parameters to the exTR-TR method's overall sensitivity. Most strikingly, uncertainties in the ratio of moments (parameter (v)) seem to contribute only to a negligible extend to the overall sensitivity (panels (m), (n), (o)). Measurement uncertainties in the stratospheric mixing ratio $\chi(x)$ contribute evenly distributed to the overall

sensitivity to a moderate extend (panels (j), (k), (l)). Due to the slightly worse measurement precision during SouthTRAC and in addition due to the decelerating relative growth rate of $SF_6$ mixing ratios, the uncertainties in $\chi(x)$ have a stronger impact on the overall sensitivity during SouthTRAC than during the other two campaigns. In the upper part of the LMS (above Δθ=50 K), uncertainties in $t_{TR}$ dominate the overall sensitivity (panels (g), (h), (i)). Below, uncertainties in $t_{exTR}$ and in $f_{exTR}(x)$ gain importance (panels (a)-(f)). Note that for the SH uncertainties in $t_{exTR}$ are low (see Table 1), which is

reflected by contributing only to a minor to moderate extend to the overall sensitivity during the SouthTRAC campaign (panel (f)).

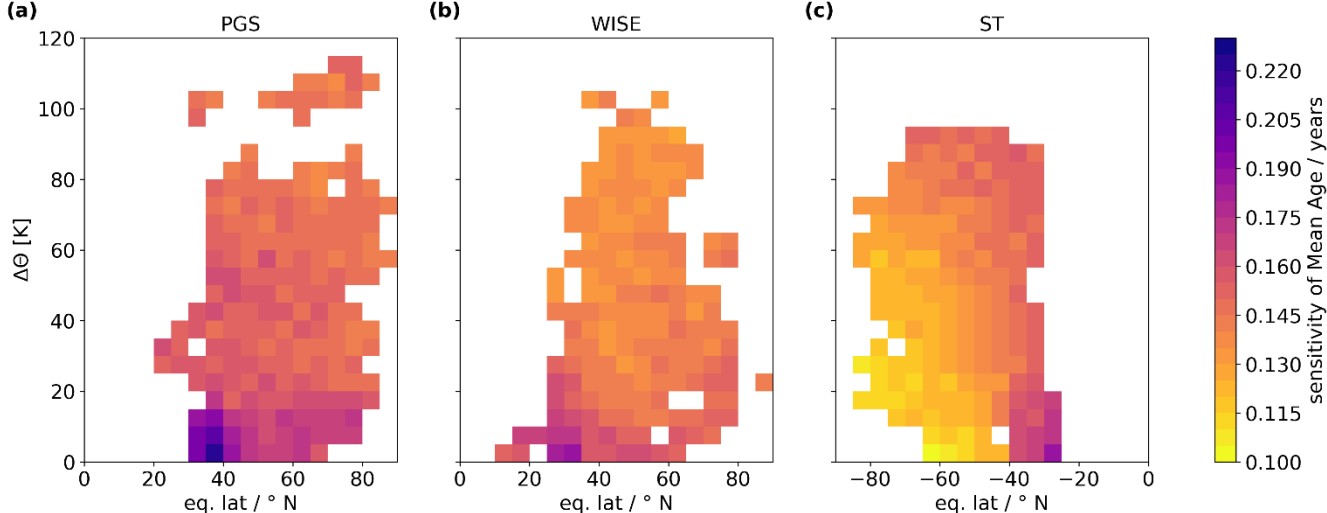

**Figure 5.** Latitude-altitude distribution of sensitivities of mean age from the exTR-TR method to uncertainties in all considered input parameters. Calculated for PGS (a), WISE (b) and SouthTRAC (c) campaign. Altitude is given in Δθ. Horizontally, data are sorted by eq. lat.



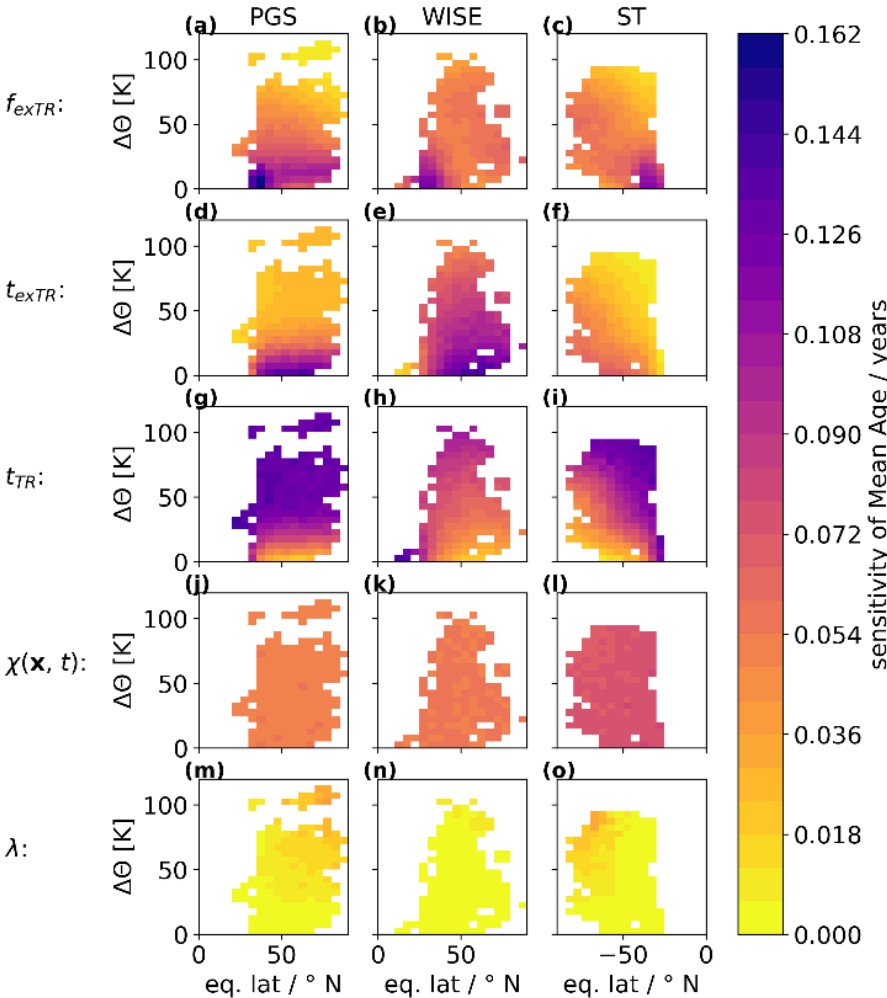

**Figure 6.** As Fig. 5, but each uncertain parameter is considered independently, assuming the others to be accurate. Uncertain parameters: exTR origin fraction ($f_{exTR}$, top row), time shift to exTR entry region ($t_{exTR}$, second row), time shift to TR entry region ($t_{TR}$, third row), SF$_6$ mixing ratio measurement uncertainty ($\chi(\mathbf{x})$, forth row) and the ratio of moments ($\lambda$, bottom row).

## 4 Summary and Conclusions

In this work, the new exTR-TR method to derive stratospheric mean age of air in the LMS from observational tracer mixing ratio data is presented. In order to take exTR input into the stratosphere into account, our implementation of the exTR-TR method uses seasonally averaged exTR origin fractions from CLaMS (Hauck et al., 2020), for which we provide a parameterization, and a long term tracer mixing ratio time series for each entry region $\chi(\boldsymbol{x}_i, t')$. Following (Hauck et al., 2020), the entry regions are defined as a northern (90-30° N), a TR (30° N - 30° S) and a southern (30-90° S) tropopause section. Owing to the lack of continuous long term measurements of age tracers at the tropopause, we approximated $\chi(\boldsymbol{x}_i, t')$ by applying a constant empirical time shift $t_{xi}$ to the available TR ground tracer mixing ratio time series for each entry



region. For the age tracer SF₆, individual $t_{xi}$ were obtained in this study by averaging optimized three dimensional model output between 1973-2008 that was published by Rigby et al. (2010). We emphasize that the resulting $t_{xi}$ are exclusively

used to approximate $\chi(x_i, t')$ and that they do not represent real-world transport times between TR ground and the entry regions.

We applied the exTR-TR method to in situ SF₆ measurements taken during three aircraft campaigns in different geographical regions and at different times: PGS in NH late winter/early spring 2016, WISE in NH fall 2017 and SouthTRAC in SH early spring 2019. The resulting mean age values were averaged into bins over multiple flights using tropopause-relative altitude

and eq. lat. coordinates (Δθ-bins of size 5 K, eq. lat. bins of size 5°). These averaged mean age values were corrected for mesospheric loss by applying a linear function published by Leedham Elvidge et al. (2018). In addition, the conventional TR-only method, which assumes that all air enters the stratosphere through the TR tropopause, was applied to the same data and the results were post-processed in the same way in order to compare the results. Using the conventional TR-only method negative mean age values are derived in the LMS close to the tropopause during the WISE campaign. Using the new exTR-

TR method instead, the number and extend of negative mean age values is reduced. Maximum absolute differences between the resulting averaged mean age values from the two methods range from 0.31 years to 0.42 years among the three campaigns and go in different directions for the two hemispheres. With our new exTR-TR method we are thus able to derive more realistic values of typical transport times in the LMS from measurements. This allows comparison of the two hemispheres based on campaign data. We compared results derived using the exTR-TR method from PGS campaign phase 2

(PGS2) to SouthTRAC campaign phase 1 (ST1) in order to investigate hemispheric differences with a maximal seasonal overlap of the campaigns. On average, below Δθ=65 K the LMS was 0.5 years ± 0.3 years older during PGS2 than during ST1 across all eq. lats that are covered by both datasets. We attribute this older LMS to mixing with old vortex air during PGS2, opposed to a more confined vortex edge with higher age gradients during ST1. Although these findings only cover an isolated time period of less than two months for each campaign and do not account for different polar vortex characteristics,

they are in agreement with multiannual simulation results from Konopka et al. (2015), who found a pronounced minimum in wave forcing driving the shallow branch of the BDC in the midlatitudes of the lower stratosphere in the SH between June and October, opposed to a maximum in boreal spring in the NH.

The sensitivity of the exTR-TR method to uncertainties of six input parameters was investigated at different locations using a Monte Carlo approach. The mesospheric loss of SF₆ was excluded from this sensitivity analysis, since it is currently not well

understood and beyond the scope of this work. The combined sensitivity was found to be less than 0.22 years for all locations for all three campaigns. The most sensitive region for each hemisphere was located between 20-40 ° poleward of the equator below Δθ=20 K. This is related to the setup of the experiment with a boundary at 30° in each hemisphere. Uncertainties in the origin fractions and in $t_{xi}$ have the largest isolated impact on the sensitivity of the exTR-TR method. Overall, these sensitivities are lower than the differences between mean ages derived using the exTR-TR method and the

TR-only method. Hence, our new exTR-TR method yields mean age values that differ considerably from results obtained using the conventional TR-only method in the LMS. In future studies, the exTR-TR method could be used to improve



deriving estimates of total and inorganic chlorine from observations of organic chlorine in the LMS like in Jesswein et al. (2021).

**Appendix A: Calculating mean age in the LMS considering multiple entry regions and an ideal tracer**

In case of an ideal inert linear evolving tracer, the tropical ground time series as a function of transit time $t'$ is given by

$$\chi(x_{TR\ ground}, t') = a - bt'. \tag{A1}$$

The negative sign points out, that looking at increasing transit times means looking backwards in time.

Assuming a constant time shift $t_{xi}$ for each entry region i, the tracer time series at $x_i$ is

$$\chi(x_i, t') = a - b * (t' - t_{xi}). \tag{A2}$$

Hence, by inserting Eq. (A2) into Eq. (2), the stratospheric mixing ratio can be expressed as

$$\chi(x) = \int_0^\infty \sum_{i=0}^{N-1} \left( f_i(x) * \left( a - b * (t' - t_{xi}) \right) \right) * G(x, t')dt'. \tag{A3}$$

Since the sum of all origin fractions equals 1, Eq. (A3) can also be written as

$$\chi(x) = \int_0^\infty [a - bt' + b * \sum_{i=0}^{N-1}(f_i(x) * t_{xi})] * G(x, t')dt', \tag{A4}$$

which is equivalent to

$$\chi(x) = a + b * \sum_{i=0}^{N-1}(f_i(x) * t_{xi}) - b * \int_0^\infty t' * G(x, t')dt'. \tag{A5}$$

The mean age $\Gamma$ is the first moment of the age spectrum, given by

$$\Gamma(x) = \int_0^\infty t' * G(x, t')dt'. \tag{A6}$$

Inserting Eq. (A6) into Eq. (A5) yields:

$$\chi(x) = a + b * \sum_{i=0}^{N-1}(f_i(x) * t_{xi}) - b * \Gamma(x). \tag{A7}$$

Equation (A7) can be solved for $\Gamma$, which yields

$$\Gamma(x) = \frac{a - \chi(x)}{b} + \sum_{i=0}^{N-1}(f_i(x) * t_{xi}), \tag{A8}$$

which is equivalent to Eq. (5).

In order to derive mean age from an ideal inert quadratic evolving tracer with multiple entry regions, we extended the equations given by (Volk et al., 1997). In this case the TR ground mixing ratio time series is given as a function of transit

time by

$$\chi(x_{TR\ ground}, t') = a - bt' + ct'^2. \tag{A9}$$

Assuming a constant time shift $t_{xi}$ for each entry region i, the tracer time series at $x_i$ is

$$\chi(x_i, t') = a - b * (t' - t_{xi}) + c * (t' - t_{xi})^2. \tag{A10}$$

Hence, by inserting Eq. (A10) into Eq. (2), the stratospheric mixing ratio can be expressed as

$$\chi(x) = \int_0^\infty \sum_{i=0}^{N-1} \left( f_i * (a - b * (t' - t_{xi}) + c * (t' - t_{xi})^2) \right) * G(x, t')dt', \tag{A11}$$

which is equivalent to





$$\chi(\boldsymbol{x}) = \sum_{i=0}^{N-1}\left(f_i * \int_0^\infty (a - b*(t'-t_{xi}) + c*(t'-t_{xi})^2) * G(\boldsymbol{x},t')dt'\right). \tag{A12}$$

Note that for better readability $f_i(\boldsymbol{x})$ is written as $f_i$.

By extracting constant factors from the integral and applying Eq. (A6), Eq. (A12) can also be written as

$$\chi(\boldsymbol{x}) = \sum_{i=0}^{N-1}\left[f_i * \left(a + bt_{xi} - b*\int_0^\infty t'*G(\boldsymbol{x},t')dt' + c*\int_0^\infty (t'-t_{xi})^2 * G(\boldsymbol{x},t')dt'\right)\right]$$

$$= \sum_{i=0}^{N-1}\left[f_i * \left(a + bt_{xi} - b\Gamma(\boldsymbol{x}) + c*\int_0^\infty (t'-t_{xi})^2 * G(\boldsymbol{x},t')dt'\right)\right]$$

$$= \sum_{i=0}^{N-1}\left[f_i * \left(a + bt_{xi} - b\Gamma(\boldsymbol{x}) + c*\int_0^\infty (t'^2 - 2t_{xi}t' + t_{xi}^2) * G(\boldsymbol{x},t')dt'\right)\right]$$

$$= \sum_{i=0}^{N-1}\left[f_i * \left(a + bt_{xi} - b\Gamma(\boldsymbol{x}) + c*\left(t_{xi}^2 + \int_0^\infty (t'^2 - 2t_{xi}t') * G(\boldsymbol{x},t')dt'\right)\right)\right]$$

$$= \sum_{i=0}^{N-1}\left[f_i * \left(a + bt_{xi} - b\Gamma(\boldsymbol{x}) + c*\left(t_{xi}^2 - 2t_{xi}\Gamma(\boldsymbol{x}) + \int_0^\infty t'^2 * G(\boldsymbol{x},t')dt'\right)\right)\right]. \tag{A13}$$

The width of the age spectrum $\Delta$ is the square root of the second centred moment of the age spectrum, which is given by

$$\Delta^2(\boldsymbol{x}) = \frac{1}{2}\int_0^\infty (t' - \Gamma(\boldsymbol{x}))^2 * G(\boldsymbol{x},t') * dt'. \tag{A14}$$

Equation (A14) can be transformed to

$$\int_0^\infty t'^2 * G(\boldsymbol{x},t')dt' = 2\Delta(\boldsymbol{x})^2 + \Gamma(\boldsymbol{x})^2. \tag{A15}$$

Inserting Eq. (A15) into Eq. (A13) yields

$$\chi(\boldsymbol{x}) = \sum_{i=0}^{N-1}\left[f_i * \left(a + bt_{xi} - b\Gamma(\boldsymbol{x}) + c*(t_{xi}^2 - 2t_{xi}\Gamma(\boldsymbol{x}) + \Gamma(\boldsymbol{x})^2 + 2\Delta(\boldsymbol{x})^2)\right)\right]. \tag{A16}$$

Since the sum of all origin fractions equals 1 and with the weighted mean time shift $t_m(\boldsymbol{x}) = \sum_{i=0}^{N-1}[f_i(\boldsymbol{x}) * t_{xi}]$, Eq. (A16) can also be written as:

$$\chi(\boldsymbol{x}) = a - b\Gamma(\boldsymbol{x}) + c*(2\Delta(\boldsymbol{x})^2 + \Gamma(\boldsymbol{x})^2) + \sum_{i=0}^{N-1}\left[f_i * \left(bt_{xi} + c*\left(t_{xi}^2 - 2t_{xi}\Gamma(\boldsymbol{x})\right)\right)\right]$$

$$= a + b*(t_m(\boldsymbol{x}) - \Gamma(\boldsymbol{x})) + c*\left(2\Delta(\boldsymbol{x})^2 + \Gamma(\boldsymbol{x})^2 + \sum_{i=0}^{N-1}\left[f_i * \left(t_{xi}^2 - 2t_{xi}\Gamma(\boldsymbol{x})\right)\right]\right)$$

$$= a + b*(t_m(\boldsymbol{x}) - \Gamma(\boldsymbol{x})) + c*(2\Delta(\boldsymbol{x})^2 + \Gamma(\boldsymbol{x})^2 - 2t_m(\boldsymbol{x})\Gamma(\boldsymbol{x}) + \sum_{i=0}^{N-1}[f_i t_{xi}^2]) \tag{A17}$$

Inserting the ratio of moments $\lambda = \Delta^2/\Gamma$ into Eq. (A17) yields the quadratic equation Eq. (A18):

$$\chi(\boldsymbol{x}) = a + b*(t_m(\boldsymbol{x}) - \Gamma(\boldsymbol{x})) + c*(2\lambda\Gamma(\boldsymbol{x}) + \Gamma(\boldsymbol{x})^2 - 2t_m(\boldsymbol{x})\Gamma(\boldsymbol{x}) + \sum_{i=0}^{N-1}[f_i t_{xi}^2]), \tag{A18}$$

which can be rearranged to

$$0 = \frac{a + bt_m(\boldsymbol{x}) - \chi(\boldsymbol{x})}{c} + \sum_{i=0}^{N-1}[f_i t_{xi}^2] + \Gamma(\boldsymbol{x})*\left(2\lambda - \frac{b}{c} - 2t_m(\boldsymbol{x})\right) + \Gamma(\boldsymbol{x})^2, \tag{A19}$$

and finally solved for $\Gamma$:

$$\Gamma(\boldsymbol{x})_{1,2} = -\lambda + t_m + \frac{b}{2c} \pm \sqrt{\left(-\lambda + t_m + \frac{b}{2c}\right)^2 - \frac{a + bt_m - \chi(\boldsymbol{x})}{c} - \sum_{i=0}^{N-1}[f_i * t_{xi}^2]}. \tag{A21}$$



**Appendix B: CLaMS origin fraction parameterizations**

We designed a general mathematical parameterization function $\varphi_{i,seas}$ with 12 parameters to derive 2-D parameterizations for exTR origin fractions in $\Delta\Theta - eq.\,lat$ space. The process of designing $\varphi_{i,seas}$ was guided by a non-physical but entirely

geometrical approach pursuing three priorities for all 10 considered $f_{i,seas}(\boldsymbol{x})$ at once:

(i) $\varphi_{i,seas}$ should be able to reproduce major geometrical features of the distributions.

(ii) The maximum difference to $f_{i,seas}(\boldsymbol{x})$ should be as low as possible.

(iii) The mean deviation to $f_{i,seas}(\boldsymbol{x})$ should be as low as possible.

In addition to the three priorities, the number of parameters needed to achieve (i), (ii) and (iii) should be preferably low. The

resulting general mathematical parameterization function is a combination of Gaussian distributions and cumulative Gumbel distributions with 12 parameters in total:

$$\text{peak1}(eq.\,lat,\Delta\Theta) = e^{-e^{-\frac{|eq.lat|-x_0}{x_1}}} * e^{-\left(\frac{\Delta\Theta-y_1}{y_0}\right)^2} \tag{B1}$$

$$\text{peak2}(eq.\,lat,\Delta\Theta) = g_a * e^{-\left(\frac{eq.lat-g_{x1}}{g_{x0}}\right)^2} * e^{-\left(\frac{\Delta\Theta-g_{y1}}{g_{y0}}\right)^2} \tag{B2}$$

$$\text{offset\_gumbel}(\Delta\Theta) = b_y * e^{-e^{-\frac{\Delta\Theta-e_0}{e_1}}} \tag{B3}$$

$$\varphi_{i,seas}(eq.\,lat,\Delta\Theta) = \text{peak1}(eq.\,lat,\Delta\Theta) + \text{peak2}(eq.\,lat,\Delta\Theta) + \text{offset\_gumbel}(\Delta\Theta) \tag{B4}$$

The seasonally averaged $f_{i,seas}(\boldsymbol{x})$ data published by (Hauck et al., 2020) is gridded in 2° latitude and 37 vertical potential temperature levels between 280 K and 3000 K. Additionally, the potential temperature difference to the local tropopause ($\Delta\Theta$) is provided for each data point. In order to find optimal fitting parameters using a least-square fit, for each of the 10 considered $f_{i,seas}(\boldsymbol{x})$ we only considered data from the respective hemisphere and for the lower 20 vertical levels (i.e. 280 K

to 480 K). The resulting parameters for each of the 10 considered $f_{i,seas}(\boldsymbol{x})$ are listed in Table B1. The python code for applying $\varphi_{i,seas}$ as given in Eq. (B4) and automatically including the information given in Table B1 is available from Wagenhäuser (2022a).

**Table B1:** Parameter values for extra tropical origin fractions $\boldsymbol{\varphi_{i,seas}}$ calculated with Eq. (B4) by hemisphere and season.

| | $x_0$ | $x_1$ | $y_0$ | $y_1$ | $b_y$ | $e_0$ | $e_1$ | $g_{y0}$ | $g_{y1}$ | $g_a$ | $g_{x0}$ | $g_{x1}$ |
|---|---|---|---|---|---|---|---|---|---|---|---|---|
| NH ANN | 31.012 | 4.873 | 33.943 | -17.027 | 0.132 | 13.288 | 13.039 | 63.375 | 43.149 | 0.228 | 50.778 | 61.048 |
| NH DJF | 31.581 | 5.507 | 52.037 | -20.088 | 0.107 | 36.32 | 9.341 | 77.047 | 82.829 | 0.132 | 73.343 | 50.228 |
| NH MAM | 31.786 | 3.858 | 33.519 | -16.758 | -669.405 | -41.6 | 1.56 | 11043.854 | 85.872 | 669.578 | 6003.475 | 80.459 |
| NH JJA | 36.242 | 18.45 | 17.177 | -5.66 | 0.142 | 10.948 | 4.301 | -49.379 | 28.372 | 0.308 | 40.474 | 48.496 |
| NH SON | 26.958 | 8.66 | 40.786 | -19.772 | 0.152 | 17.019 | 8.001 | -53.255 | 52.041 | 0.366 | 58.629 | 59.063 |
| SH ANN | 30.434 | 4.033 | -32.442 | -15.257 | 0.13 | 75.493 | 12.132 | 41.952 | 44.968 | 0.488 | 66.713 | -86.659 |
| SH DJF | 29.478 | 7.284 | -48.731 | -31.197 | 0.088 | 327.402 | -3.53 | 59.583 | 49.447 | 0.355 | 59.492 | -87.584 |



| SH MAM | 29.776 | 5.449 | -44.403 | -21.513 | 0.12 | 25.627 | 10.129 | 57.057 | 49.173 | 0.253 | 52.374 | -59.583 |
| SH JJA | 30.733 | 3.882 | -26.612 | -9.583 | 0.136 | 67.734 | 8.593 | 31.786 | 43.443 | 0.381 | 58.274 | -80.771 |
| SH SON | 30.296 | 4.814 | -36.864 | -14.732 | 0.135 | 30.789 | 49.448 | 37.568 | 45.222 | 0.585 | 36.574 | -85.755 |


## Appendix C: SouthTRAC phase 1 and PGS phase 2 campaign differences using the TR-only method

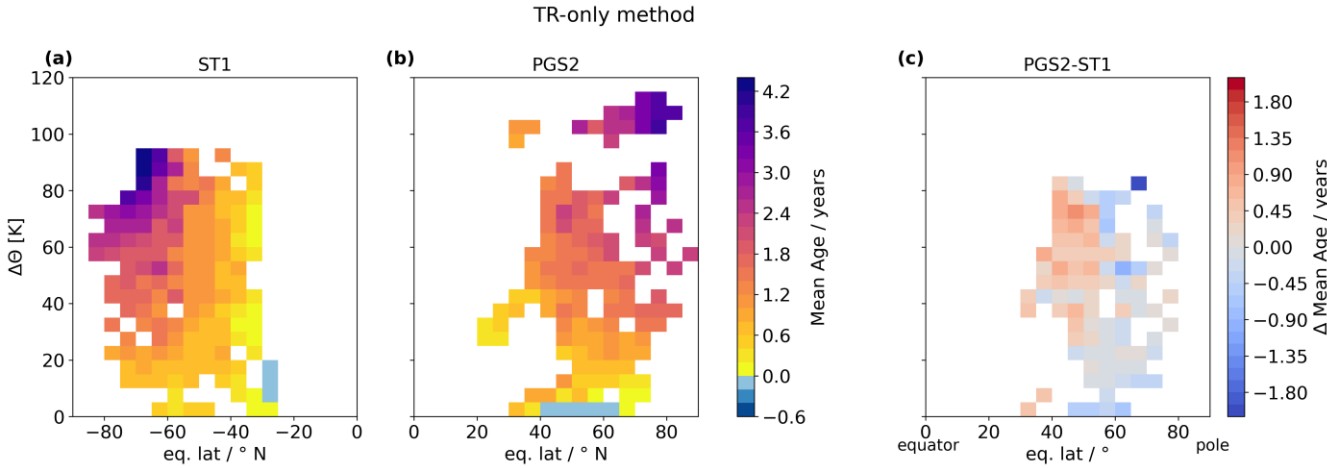

**Figure C1:** Same as Fig. 4, but mean ages have been derived using the conventional TR-only method instead.

## Code and data availability

The Python software implementation of the exTR-TR method is available from Wagenhäuser and Engel (2022).

The Python software code repository "f_exTR" for deploying our parameterizations of the CLaMS origin fractions is available from Wagenhäuser (2022a).

The Python software code repository "sf6-timeshifts-from-rigby2010" for deriving $SF_6$ time shifts to tropical ground using model data from Rigby et al. (2010) (Sect. 2.2.2) is available from Wagenhäuser (2022b).

Tracer measurements, flight coordinates and mean age values derived using both the exTR-TR method and the TR-only method can be downloaded from Wagenhäuser et al. (2022).

## Author contribution

TW developed the mathematical framework and Python software code for the exTR-TR method in close collaboration with AE. AE initiated this study. TW, MJ, TK, TS and AE operated the GhOST instrument during the SouthTRAC campaign.

TW wrote the manuscript in collaboration with AE. All authors contributed to the final version of the manuscript.



**Competing interests**

The authors declare that they have no conflict of interest.

**Acknowledgements**

This research was supported under the Deutsche Forschungsgemeinschaft (DFG, German Research Foundation) Priority
Program SPP 1294 "Atmospheric and Earth System Research with HALO" – "High Altitude and Long Range Research
Aircraft" under project numbers EN367/5, EN367/8, EN367/11, EN 367/13, EN 367/14 and EN 367/16. Financial support
also came from the DFG – TRR 301 – Project-ID 428312742. We would like to thank the DLR staff for the operation of the
HALO and the support during all three campaigns. Many thanks also to all former master students at Goethe-University
Frankfurt, who helped carrying out measurements during the campaigns. Moreover, we thank Jens-Uwe Grooß for
facilitating access to model based $\Delta\theta$ and eq. lat. data along the flight tracks. We thank Matthew Rigby and Luke Western
for providing updated AGAGE 12-box model output of $SF_6$ mixing ratios. We further thank Ronald Prinn, Ray Weiss, Paul
Krummel, Dickon Young, Simon O'Doherty and Jens Mühle for facilitating access to the AGAGE data (agage.mit.edu).
The AGAGE stations used in this paper are supported by the National Aeronautics and Space Administration (NASA)
(grants NNX16AC98G to MIT, and NNX16AC97G and NNX16AC96G to SIO). Support also comes from the UK
Department for Business, Energy & Industrial Strategy (BEIS) for MHD, the National Oceanic and Atmospheric
Administration (NOAA) for RPB, and the Commonwealth Scientific and Industrial Research Organization (CSIRO) and
the Bureau of Meteorology (Australia) for CGO.

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
