# Peer review of "Mean age from observations in the lowermost stratosphere: an improved method and interhemispheric differences"

_EGUsphere, 2022_

## Author Comment (AC2)

**Reply to RC2**

**Manuscript information:**

- Title: Mean age from observations in the lowermost stratosphere: an improved method and interhemispheric differences
- Author(s): Thomas Wagenhäuser, Markus Jesswein, Timo Keber, Tanja Schuck, and Andreas Engel
- MS No.: egusphere-2022-1197
- MS type: Research article
- Iteration: Final response

We would like to thank the reviewer for the kind words and the constructive comments. In the following document, the reviewers' comments are marked in *italic font* and indented, our answers are in regular font. Changes in the manuscript are marked-up in red and listed as framed screenshots below the respective comment. The line numbers in our listed changes refer to the marked-up version of the revised manuscript, that is provided separately.

**Point-by-Point reply**

1. *Minor Comment 1, Line 86: It is not clear to me why G(x,t') in equation (2) does not depend on the source region x_i. That is, G should also be conditionally dependent on x_i, such that G(x,t') should be rewritten as G(x,t'|x_i). The authors here have assumed that the transport operators propagating tracer concentrations for all regions i are the same, but I can envision several cases where this would not be true. For example, air propagating into the stratosphere at high latitudes will have no clear path into the stratosphere, as opposed to air straddling the midlatitude tropopause, where isentropic surfaces provide a clear pathway for stratosphere-troposphere exchange. The authors need to provide their rationale here.*

Thank you for your constructive comment. In case of an ideal inert linear evolving tracer the differences across individual $G(x,t'|x\_i)$ have no influence on calculating the mean age. In contrast, in the quadratic tracer case the mean age cannot be calculated without knowledge of $\Gamma(x|x_i)$. However, if the quadratic term of the tracer mixing ratio time series is sufficiently low, then the concept of $G(x,t'|x\_i)$ can be neglected by using the Ansatz expressed in Eq. (2). We concluded that within the scope of this study, which focuses on relatively young mean ages derived from $SF_6$, we can neglect the influence of differences between different $G(x,t'|x\_i)$.

We revised the Appendix A in the updated version of the manuscript to clarify our approach:

Appendix A: Calculating mean age in the LMS considering multiple entry regions and an ideal tracer

In case of an ideal inert linear evolving tracer, the tropical ground time series as a function of transit time $t'$ is given by

$$\chi(x_{TR\ ground}, t') = a - bt'. \tag{A1}$$

The negative sign points out, that looking at increasing transit times means looking backwards in time.

455 Assuming a constant time shift $t_{xi}$ for each entry region i, the tracer time series at $x_i$ is

$$\chi(x_i, t') = a - b * (t' - t_{xi}). \tag{A2}$$

Considering individual transit time distributions $G_i(x, t')$ for each origin fraction $f_i(x)$, the stratospheric mixing ratio $\chi(x)$ of a suitable age tracer at an arbitrary location $x$ in the stratosphere is

$$\chi(x) = \sum_{i=0}^{N-1}[f_i(x) * \int_0^\infty \chi(x_i, t') * G_i(x, t')dt'] \tag{A3}$$

460 Hence, by inserting Eq. (A2) into Eq. (A3), the stratospheric mixing ratio can be expressed as

$$\chi(x) = \sum_{i=0}^{N-1}[f_i(x) * \int_0^\infty (a - bt' + bt_{xi}) * G_i(x, t')dt']$$

$$= \sum_{i=0}^{N-1}[f_i(x) * (a + bt_{xi})] - \sum_{i=0}^{N-1}[f_i(x) * b * \int_0^\infty t' * G_i(x, t')dt']. \tag{A4}$$

$$\chi(x) = \int_0^\infty \sum_{i=0}^{N-1}\left(f_i(x) * \left(a - b * (t' - t_{xi})\right)\right) * G(x, t')dt'. \tag{A3}$$

The mean age $\Gamma$ is the first moment of the age spectrum, given by

$$\Gamma(\boldsymbol{x}) = \int_0^\infty t' * G(\boldsymbol{x}, t')dt'. \hspace{4cm} (A5)$$

In case of $G_i(\boldsymbol{x}, t')$ Eq. (A5) translates into the mean age of air originating from source region i ($\Gamma_i(\boldsymbol{x})$)

$$\Gamma_i(\boldsymbol{x}) = \int_0^\infty t' * G_i(\boldsymbol{x}, t')dt'. \hspace{4cm} (A6)$$

Inserting Eq. (A6) into Eq. (A4) yields:

$$\chi(\boldsymbol{x}) = \sum_{i=0}^{N-1}[f_i(\boldsymbol{x}) * (a + bt_{xi})] - \sum_{i=0}^{N-1}[f_i(\boldsymbol{x}) * b * \Gamma_i(\boldsymbol{x})]. \hspace{2cm} (A7)$$

Since the sum of all origin fractions equals 1, Eq. (A7) can also be written as

$$\chi(\boldsymbol{x}) = a + \sum_{i=0}^{N-1}[f_i(\boldsymbol{x}) * bt_{xi}] - b * \sum_{i=0}^{N-1}[f_i(\boldsymbol{x}) * \Gamma_i(\boldsymbol{x})]. \hspace{2cm} (A8)$$

The mean age $\Gamma(\boldsymbol{x})$ equals the sum of individual $\Gamma_i(\boldsymbol{x})$, weighted by their respective origin fraction $f_i(\boldsymbol{x})$:

$$\Gamma(\boldsymbol{x}) = \sum_{i=0}^{N-1}[f_i(\boldsymbol{x}) * \Gamma_i(\boldsymbol{x})]. \hspace{4cm} (A9)$$

By inserting Eq. (A9) into Eq. (A8), we can thus reduce the number of unknown parameters:

$$\chi(\boldsymbol{x}) = a + \sum_{i=0}^{N-1}[f_i(\boldsymbol{x}) * bt_{xi}] - b * \Gamma(\boldsymbol{x}) \hspace{3cm} (A10)$$
* * *
$$\chi(\boldsymbol{x}) = \int_0^\infty [a - bt' + b * \sum_{i=0}^{N-1}(f_i(\boldsymbol{x}) * t_{xi})] * G(\boldsymbol{x}, t')dt', \hspace{2cm} (A4)$$

which is equivalent to

$$\chi(\boldsymbol{x}) = a + b * \sum_{i=0}^{N-1}(f_i(\boldsymbol{x}) * t_{xi}) - b * \int_0^\infty t' * G(\boldsymbol{x}, t')dt'. \hspace{2cm} (A5)$$

The mean age $\Gamma$ is the first moment of the age spectrum, given by

$$\Gamma(\boldsymbol{x}) = \int_0^\infty t' * G(\boldsymbol{x}, t')dt'. \hspace{4cm} (A6)$$

Inserting Eq. (A6) into Eq. (A5) yields:

$$\chi(\boldsymbol{x}) = a + b * \sum_{i=0}^{N-1}(f_i(\boldsymbol{x}) * t_{xi}) - b * \Gamma(\boldsymbol{x}). \hspace{3cm} (A7)$$

Equation (A10) can be solved for $\Gamma$, which yields

$$\Gamma(\boldsymbol{x}) = \frac{a - \chi(\boldsymbol{x})}{b} + \sum_{i=0}^{N-1}(f_i(\boldsymbol{x}) * t_{xi}), \hspace{3cm} (A9)$$
* * *
which is equivalent to Eq. (5). The same result can be obtained mathematically when we use the origin fractions as weights only for the mixing ratio time series and neglect the concept of $G_i(\boldsymbol{x}, t')$ (starting with Eq. (2) instead of Eq. (A3)). Differences across individual $G_i(\boldsymbol{x}, t')$ thus have no influence on calculating the mean age from an ideal inert linear evolving tracer. In contrast, in case of an ideal inert quadratic evolving tracer the Ansatz expressed in Eq. (A3) cannot be solved for $\Gamma(\boldsymbol{x})$ without knowledge of individual $\Gamma_i(\boldsymbol{x})$. However, if the quadratic term of the tracer mixing ratio time series is sufficiently low, then the concept of $G_i(\boldsymbol{x}, t')$ can be neglected by using the Ansatz expressed in Eq. (2).

In order to derive mean age from an ideal inert quadratic evolving tracer with multiple entry regions, we extended the equations given by (Volk et al., 1997). In this case the TR ground mixing ratio time series is given as a function of transit time by

$$\chi(\boldsymbol{x}_{TR\,ground}, t') = a - bt' + ct'^2. \hspace{4cm} (A9)$$
* * *
2. *Minor Comment 2, Section 2.2.2: I am curious about the calculation of t_xi. The procedure outlined in steps (i)-(iii) essentially sounds like a description of how to calculate the SF6-age, which previous studies have used to calculate the tropospheric mean age (albeit using an SF6 surface boundary condition that only averages stations over northern midlatitudes). The details of the regions considered may be slightly different, but the procedure is basically the same. So why not reference this literature? In particular, the authors should review these studies:*

   - *Waugh, Darryn W., A. M. Crotwell, E. J. Dlugokencky, G. S. Dutton, J. W. Elkins, B. D. Hall, E. J. Hintsa et al. "Tropospheric SF6: Age of air from the Northern Hemisphere midlatitude surface." Journal of Geophysical Research: Atmospheres 118, no. 19 (2013): 11-429.*
   - *Orbe, Clara, Darryn W. Waugh, Stephen Montzka, Edward J. Dlugokencky, Susan Strahan, Stephen D. Steenrod, Sarah Strode et al. "Tropospheric Age of Air: Influence*

*of SF6 Emissions on Recent Surface Trends and Model Biases." Journal of Geophysical Research: Atmospheres 126, no. 19 (2021): e2021JD035451.*

Thank you for your constructive comment. We referenced the proposed literature in the updated version of the manuscript:

the subsequent decade from 2008 on. This decade is not covered by the model from Rigby et al. (2010) that we used to derive $t_{xi}$, however it is covered by $\chi(x_{TR\ ground}, t')$ (Laube et al., 2022, updated from Simmonds et al., 2020).

205 Previous studies used a similar procedure as outlined above (steps (i) to (iii)) to estimate transport time scales while referencing the NH midlatitude ground (Orbe et al., 2021; Waugh et al., 2013). We found that $t_{xi}$ varies less over the time period 1973 to 2008 when referencing the tropical ground in the MOZART data set. In order to derive more robust entry mixing ratio time series for our exTR-TR method, we thus decided to use the tropical ground as a reference. We emphasize that each $t_{xi}$ as defined here is an integrated empirical measure. $t_{xi}$ does neither contain useful information on transport

210 paths nor on transit times from the TR ground to the entry regions. We only use $t_{xi}$ to derive entry mixing ratio time series at locations, where suitable long term time series are not available from measurements.

3. *Technical Comments: Line 83: The concept of "origin fraction" referred to here certainly precedes the Hauck et al. (2020) study and the authors should properly reference the literature. For example, see these studies:*

   - *Orbe, Clara, Mark Holzer, Lorenzo M. Polvani, and Darryn Waugh. "Airmass origin as a diagnostic of tropospheric transport." Journal of Geophysical Research: Atmospheres 118, no. 3 (2013): 1459-1470.*
   - *Orbe, Clara, Darryn W. Waugh, and Paul A. Newman. "Airmass origin in the tropical lower stratosphere: The influence of Asian boundary layer air." Geophysical Research Letters 42, no. 10 (2015): 4240-4248.*

Thank you for pointing that out. We referenced the proposed studies in the updated version of our manuscript:

$\chi(x_i, t')$ by calculating a weighted mixing ratio time series. The relative importance of individual source regions can be described by so-called origin fractions (e.g. Orbe et al., 2013, 2015). We use the origin fractions $f_i(x)$ as  derived

85 by Hauck et al. (2020) as weights for each $\chi(x_i)$. $f_i(x)$ is the fraction of air at $x$, that entered the stratosphere through $x_i$. By

---

## Author Response (AR1)

**Author's response**

**Manuscript information:**

- Title: Mean age from observations in the lowermost stratosphere: an improved method and interhemispheric differences
- Author(s): Thomas Wagenhäuser, Markus Jesswein, Timo Keber, Tanja Schuck, and Andreas Engel
- MS No.: egusphere-2022-1197
- MS type: Research article
- Iteration: Final response

We would like to thank the three reviewers for the kind words and the constructive comments. In the following document, the reviewers' comments are marked in *italic font* and indented, our answers are in regular font. Changes in the manuscript are marked-up in red and listed as framed screenshots below the respective comment. The line numbers in our listed changes refer to the marked-up version of the revised manuscript, that is provided separately.

**Reply to RC1**

**Point-by-Point reply**

1. *Line 192: add 'of' after 'instead'*

Done.

> 190 with $SF_6$ source regions being located primarily in the northern hemisphere (Rigby et al., 2010).
>
> We performed a Monte-Carlo simulation in order to test if $t_{xi}$ can be considered to be constant over time for each entry region. Firstly, for each entry region we calculated weighted means and standard deviations for each year (instead of for the

2. *Table 2 could go in the supplement.*

Done, thanks. We applied small changes to the manuscript accordingly:

> 195 create 10000 time series for each entry region. Thirdly, we applied a linear fit to each of the 10000 time series and calculated the mean and the standard deviation of the slope for each entry region. The resulting mean slopes, standard deviations and the ratio of mean slope and standard deviation are listed in Table S1 in the supplementary information Table 2. For NH exTR

> **Table 2:** Mean slopes and slope standard deviation from Monte-Carlo simulation using the data shown in Fig. 2. Ratios of mean slopes
> 215 and standard deviations have been calculated prior to rounding.
>
> | | mean time shift slope / years years-1 | time shift slope standard deviation / years years-1 | ratio of mean slope and standard deviation | |
> |---|---|---|---|---|
> | NH exTR entry region | $-2 * 10^{-3}$ | $3 * 10^{-3}$ | $-0.9$ | |
> | TR entry region | $-5 * 10^{-4}$ | $3 * 10^{-3}$ | $-0.2$ | |
> | SH exTR entry region | $-1 * 10^{-3}$ | $2 * 10^{-3}$ | $-1.1$ | |

We updated the supplementary information accordingly:

|  | mean time shift slope / years years-1 | time shift slope standard deviation / years years-1 | ratio of mean slope and standard deviation |
|---|---|---|---|
| NH exTR entry region | $-2 * 10^{-3}$ | $3 * 10^{-3}$ | $-0.9$ |
| TR entry region | $-5 * 10^{-4}$ | $3 * 10^{-3}$ | $-0.2$ |
| SH exTR entry region | $-1 * 10^{-3}$ | $2 * 10^{-3}$ | $-1.1$ |

*3. Line 246: 'datasets were processed in three steps.'*

Done.

245 dynamical tropopause (defined by the value of 2 PVU) Δθ is used as vertical coordinate. Horizontally, data are sorted by eq. lat. In order to visualize and compare our results, datasets were processed in a three-steps process:

*4. Line 250: I don't see this age correction formula in the Leedham Elvidge et al. paper. How was this derived?*

We took the correction function that is shown in Fig. 4 of the Leedham Elvidge et al. paper. There, linear fits for different subsets of their data are given in the legend. We took the top one, named "All (no tropical)". We refer to this Fig. 4 in the revised version of our manuscript:

250    3. → The averaged mean ages have been corrected for mesospheric loss using a linear correction function by (Leedham Elvidge et al., (2018), given in their Fig. 4:

$$\Gamma_{corr} = 0.85 * \Gamma - 0.02 \; years \qquad\qquad (7)$$

*5. Lines 313, 385, 390, 420: 'extend' should be 'extent'.*

Done.

310 mixed vertically and horizontally with young air in the LMS. The vortex edge is less sharp than during ST1, resulting in younger air at high latitudes and altitudes and older air outside the PGS2 vortex region compared to ST1. ¶
These results cover only isolated time periods of less than two months for each campaign. In addition, as discussed by Jesswein et al. (2021) the extent of the respective polar vortices and therefore also the location of the respective vortex edge

380 The contribution of the individual parameters (i)-(v) is shown in Fig. 6. Each row depicts isolated sensitivities to uncertainties in a single parameter with all other parameters being held at their best estimate. This allows us to test the relative importance of the individual parameters to the exTR-TR method's overall sensitivity. Most strikingly, uncertainties in the ratio of moments (parameter (v)) seem to contribute only to a negligible extent to the overall sensitivity (panels (m),

390 in the ratio of moments (parameter (v)) seem to contribute only to a negligible extent to the overall sensitivity (panels (m), (n), (o)). Measurement uncertainties in the stratospheric mixing ratio $\chi(x)$ contribute evenly distributed to the overall sensitivity to a moderate extent (panels (j), (k), (l)). Due to the slightly worse measurement precision during

**Reply to RC2**

**Point-by-Point reply**

6. *Minor Comment 1, Line 86: It is not clear to me why G(x,t') in equation (2) does not depend on the source region x_i. That is, G should also be conditionally dependent on x_i, such that G(x,t') should be rewritten as G(x,t'|x_i). The authors here have assumed that the transport operators propagating tracer concentrations for all regions i are the same, but I can envision several cases where this would not be true. For example, air propagating into the stratosphere at high latitudes will have no clear path into the stratosphere, as opposed to air straddling the midlatitude tropopause, where isentropic surfaces provide a clear pathway for stratosphere-troposphere exchange. The authors need to provide their rationale here.*

Thank you for your constructive comment. In case of an ideal inert linear evolving tracer the differences across individual $G(x,t'|x\_i)$ have no influence on calculating the mean age. In contrast, in the quadratic tracer case the mean age cannot be calculated without knowledge of $\Gamma(x|x_i)$. However, if the quadratic term of the tracer mixing ratio time series is sufficiently low, then the concept of $G(x,t'|x\_i)$ can be neglected by using the Ansatz expressed in Eq. (2). We concluded that within the scope of this study, which focuses on relatively young mean ages derived from SF$_6$, we can neglect the influence of differences between different $G(x,t'|x\_i)$.

We revised the Appendix A in the updated version of the manuscript to clarify our approach:

**Appendix A: Calculating mean age in the LMS considering multiple entry regions and an ideal tracer**

In case of an ideal inert linear evolving tracer, the tropical ground time series as a function of transit time $t'$ is given by

$$\chi(x_{TR\;ground}, t') = a - bt'. \tag{A1}$$

The negative sign points out, that looking at increasing transit times means looking backwards in time.

455  Assuming a constant time shift $t_{xi}$ for each entry region i, the tracer time series at $x_i$ is

$$\chi(x_i, t') = a - b * (t' - t_{xi}). \tag{A2}$$

Considering individual transit time distributions $G_i(x, t')$ for each origin fraction $f_i(x)$, the stratospheric mixing ratio $\chi(x)$ of a suitable age tracer at an arbitrary location $x$ in the stratosphere is

$$\chi(x) = \sum_{i=0}^{N-1}\left[ f_i(x) * \int_0^\infty \chi(x_i, t') * G_i(x, t')dt'\right] \tag{A3}$$

460  Hence, by inserting Eq. (A2) into Eq. (A3), the stratospheric mixing ratio can be expressed as

$$\chi(x) = \sum_{i=0}^{N-1}\left[ f_i(x) * \int_0^\infty (a - bt' + bt_{xi}) * G_i(x, t')dt'\right]$$

$$= \sum_{i=0}^{N-1}[ f_i(x) * (a + bt_{xi})] - \sum_{i=0}^{N-1}\left[ f_i(x) * b * \int_0^\infty t' * G_i(x, t')dt'\right]. \tag{A4}$$

$$ \tag{A3}$$

The mean age $\Gamma$ is the first moment of the age spectrum, given by

$$\Gamma(x) = \int_0^\infty t' * G(x,t')dt'. \tag{A5}$$

In case of $G_i(x,t')$ Eq. (A5) translates into the mean age of air originating from source region i ($\Gamma_i(x)$)

$$\Gamma_i(x) = \int_0^\infty t' * G_i(x,t')dt'. \tag{A6}$$

Inserting Eq. (A6) into Eq. (A4) yields:

$$\chi(x) = \sum_{i=0}^{N-1}[f_i(x) * (a + bt_{xi})] - \sum_{i=0}^{N-1}[f_i(x) * b * \Gamma_i(x)]. \tag{A7}$$

Since the sum of all origin fractions equals 1, Eq. (A7) can also be written as

$$\chi(x) = a + \sum_{i=0}^{N-1}[f_i(x) * bt_{xi}] - b * \sum_{i=0}^{N-1}[f_i(x) * \Gamma_i(x)]. \tag{A8}$$

The mean age $\Gamma(x)$ equals the sum of individual $\Gamma_i(x)$, weighted by their respective origin fraction $f_i(x)$:

$$\Gamma(x) = \sum_{i=0}^{N-1}[f_i(x) * \Gamma_i(x)]. \tag{A9}$$

By inserting Eq. (A9) into Eq. (A8), we can thus reduce the number of unknown parameters:

$$\chi(x) = a + \sum_{i=0}^{N-1}[f_i(x) * bt_{xi}] - b * \Gamma(x) \tag{A10}$$
* * *
$$\chi(x) = \int_0^\infty[a - bt' + b * \sum_{i=0}^{N-1}(f_i(x) * t_{xi})] * G(x,t')dt', \tag{A4}$$

which is equivalent to

$$\chi(x) = a + b * \sum_{i=0}^{N-1}(f_i(x) * t_{xi}) - b * \int_0^\infty t' * G(x,t')dt'. \tag{A5}$$

The mean age $\Gamma$ is the first moment of the age spectrum, given by

$$\Gamma(x) = \int_0^\infty t' * G(x,t')dt'. \tag{A6}$$

Inserting Eq. (A6) into Eq. (A5) yields:

$$\chi(x) = a + b * \sum_{i=0}^{N-1}(f_i(x) * t_{xi}) - b * \Gamma(x). \tag{A7}$$

Equation (A10) can be solved for $\Gamma$, which yields

$$\Gamma(x) = \frac{a - \chi(x)}{b} + \sum_{i=0}^{N-1}(f_i(x) * t_{xi}), \tag{A9}$$
* * *
which is equivalent to Eq. (5). The same result can be obtained mathematically when we use the origin fractions as weights only for the mixing ratio time series and neglect the concept of $G_i(x,t')$ (starting with Eq. (2) instead of Eq. (A3)). Differences across individual $G_i(x,t')$ thus have no influence on calculating the mean age from an ideal inert linear evolving tracer. In contrast, in case of an ideal inert quadratic evolving tracer the Ansatz expressed in Eq. (A3) cannot be solved for $\Gamma(x)$ without knowledge of individual $\Gamma_i(x)$. However, if the quadratic term of the tracer mixing ratio time series is sufficiently low, then the concept of $G_i(x,t')$ can be neglected by using the Ansatz expressed in Eq. (2).

In order to derive mean age from an ideal inert quadratic evolving tracer with multiple entry regions, we extended the equations given by (Volk et al., 1997). In this case the TR ground mixing ratio time series is given as a function of transit time by

$$\chi(x_{TR\ ground}, t') = a - bt' + ct'^2. \tag{A9}$$
* * *
7. *Minor Comment 2, Section 2.2.2: I am curious about the calculation of t_xi. The procedure outlined in steps (i)-(iii) essentially sounds like a description of how to calculate the SF6-age, which previous studies have used to calculate the tropospheric mean age (albeit using an SF6 surface boundary condition that only averages stations over northern midlatitudes). The details of the regions considered may be slightly different, but the procedure is basically the same. So why not reference this literature? In particular, the authors should review these studies:*

   a. *Waugh, Darryn W., A. M. Crotwell, E. J. Dlugokencky, G. S. Dutton, J. W. Elkins, B. D. Hall, E. J. Hintsa et al. "Tropospheric SF6: Age of air from the Northern Hemisphere midlatitude surface." Journal of Geophysical Research: Atmospheres 118, no. 19 (2013): 11-429.*

   b. *Orbe, Clara, Darryn W. Waugh, Stephen Montzka, Edward J. Dlugokencky, Susan Strahan, Stephen D. Steenrod, Sarah Strode et al. "Tropospheric Age of Air: Influence*

*of SF6 Emissions on Recent Surface Trends and Model Biases." Journal of Geophysical Research: Atmospheres 126, no. 19 (2021): e2021JD035451.*

Thank you for your constructive comment. We referenced the proposed literature in the updated version of the manuscript:

> the subsequent decade from 2008 on. This decade is not covered by the model from Rigby et al. (2010) that we used to derive $t_{xi}$, however it is covered by $\chi(x_{TR\ ground}, t')$ (Laube et al., 2022, updated from Simmonds et al., 2020).
>
> 205 Previous studies used a similar procedure as outlined above (steps (i) to (iii)) to estimate transport time scales while referencing the NH midlatitude ground (Orbe et al., 2021; Waugh et al., 2013). We found that $t_{xi}$ varies less over the time period 1973 to 2008 when referencing the tropical ground in the MOZART data set. In order to derive more robust entry mixing ratio time series for our exTR-TR method, we thus decided to use the tropical ground as a reference. We emphasize that each $t_{xi}$ as defined here is an integrated empirical measure. $t_{xi}$ does neither contain useful information on transport
>
> 210 paths nor on transit times from the TR ground to the entry regions. We only use $t_{xi}$ to derive entry mixing ratio time series at locations, where suitable long term time series are not available from measurements.

8. *Technical Comments: Line 83: The concept of "origin fraction" referred to here certainly precedes the Hauck et al. (2020) study and the authors should properly reference the literature. For example, see these studies:*
   a. *Orbe, Clara, Mark Holzer, Lorenzo M. Polvani, and Darryn Waugh. "Airmass origin as a diagnostic of tropospheric transport." Journal of Geophysical Research: Atmospheres 118, no. 3 (2013): 1459-1470.*
   b. *Orbe, Clara, Darryn W. Waugh, and Paul A. Newman. "Airmass origin in the tropical lower stratosphere: The influence of Asian boundary layer air." Geophysical Research Letters 42, no. 10 (2015): 4240-4248.*

Thank you for pointing that out. We referenced the proposed studies in the updated version of our manuscript:

> $\chi(x_i, t')$ by calculating a weighted mixing ratio time series. The relative importance of individual source regions can be described by so-called origin fractions (e.g. Orbe et al., 2013, 2015). We use the origin fractions $f_i(x)$ as  derived
>
> 85 by Hauck et al. (2020) as weights for each $\chi(x_i)$. $f_i(x)$ is the fraction of air at $x$, that entered the stratosphere through $x_i$. By

**Reply to RC3**

**Point-by-Point reply**

1. *Line 25. … isentrope, **and** approximates*

Done.

> The lowermost stratosphere (LMS) is the lowest part of the extra tropical (exTR) stratosphere. Its upper boundary usually is
>
> 25 defined as the 380 K isentrope,  and approximates the lower boundary of the stratosphere in the tropics. The chemical

2. *Line 39. … made contact **with***

Done.

> 40 infinitesimal fluid elements enter the stratosphere across a source region. The transit time (or "age") of each individual fluid element is the elapsed time since it last made contact  with a source region. A macroscopic air parcel in the stratosphere consists of an infinite number of such fluid elements, each with its own transit time. The transit time distribution for the air

**3. Line 48. making **fewer** assumptions **compared to** deriving age spectra.**

Done.

> 45 studies to derive the mean age of air from observations (e.g. Engel et al., 2017; Leedham Elvidge et al., 2018). Age of air from observations provides a stringent test for numerical models. The number of available trace gas observations that are suited to derive mean age is vastly higher than that to derive age spectra. In addition, deriving mean age relies on making  fewer assumptions compared to deriving age spectra. This makes mean age a valuable measure to

**4. Line 51. … measurements, an infinite lifetime is **commonly** assumed.**

Done.

> 50 depending on the trace gas used, which significantly add to the uncertainty in mean age across large areas of the stratosphere. For example, in order to derive mean age from $SF_6$ measurements,  an infinite lifetime is commonly assumed. In

**5. Line 308. Our findings indicate that… (no comma needed after 'indicate')**

Done.

> 315 Our findings indicate that on the one hand, during ST1 old air from higher altitudes descends in a confined way at high latitudes. There is a sharp vortex edge with a strong gradient in the SH. On the other hand, during PGS2 descending old air is

**6. Lines 313, 385, 390, 420: 'extend' should be 'extent'.**

Done.

> 310 mixed vertically and horizontally with young air in the LMS. The vortex edge is less sharp than during ST1, resulting in younger air at high latitudes and altitudes and older air outside the PGS2 vortex region compared to ST1. ¶
> These results cover only isolated time periods of less than two months for each campaign. In addition, as discussed by Jesswein et al. (2021) the extent of the respective polar vortices and therefore also the location of the respective vortex edge

> 380 The contribution of the individual parameters (i)-(v) is shown in Fig. 6. Each row depicts isolated sensitivities to uncertainties in a single parameter with all other parameters being held at their best estimate. This allows us to test the relative importance of the individual parameters to the exTR-TR method's overall sensitivity. Most strikingly, uncertainties in the ratio of moments (parameter (v)) seem to contribute only to a negligible extent to the overall sensitivity (panels (m),

> 390 in the ratio of moments (parameter (v)) seem to contribute only to a negligible extent to the overall sensitivity (panels (m), (n), (o)). Measurement uncertainties in the stratospheric mixing ratio $\chi(x)$ contribute evenly distributed to the overall sensitivity to a moderate extent (panels (j), (k), (l)). Due to the slightly worse measurement precision during

> 420 TR method instead, the number and extent of negative mean age values is reduced. Maximum absolute differences between